# Slicing Wasserstein over Wasserstein via Functional Optimal Transport

**Moritz Piening and Robert Beinert**
Institut für Mathematik, Technische Universität Berlin, Germany
`{piening, beinert}@tu-berlin.de`

## Abstract

Wasserstein distances define a metric between probability measures on arbitrary metric spaces, including *meta-measures* (measures over measures). The resulting *Wasserstein over Wasserstein* (WoW) distance is a powerful, but computationally costly tool for comparing datasets or distributions over images and shapes. Existing sliced WoW accelerations rely on parametric meta-measures or the existence of high-order moments, leading to numerical instability. As an alternative, we propose to leverage the isometry between the 1d Wasserstein space and the quantile functions in the function space $L_2([0,1])$. For this purpose, we introduce a general sliced Wasserstein framework for arbitrary Banach spaces. Due to the 1d Wasserstein isometry, this framework defines a sliced distance between 1d meta-measures via infinite-dimensional $L_2$-projections, parametrized by Gaussian processes. Combining this 1d construction with classical integration over the Euclidean unit sphere yields the *double-sliced Wasserstein* (DSW) metric for general meta-measures. We show that DSW minimization is equivalent to WoW minimization for discretized meta-measures, while avoiding unstable higher-order moments and computational savings. Numerical experiments on datasets, shapes, and images validate DSW as a scalable substitute for the WoW distance.

## 1 Introduction

Optimal transport (OT) enables geometrically meaningful Wasserstein distances between probability measures on arbitrary Polish spaces $\mathcal{X}$ (Villani, 2003), while remaining computationally tractable. This has had a profound impact on machine learning, where Wasserstein distances are used to train neural networks (Tong et al., 2024) and to compare data distributions (Yang et al., 2019). A key feature of OT is its applicability to non-Euclidean spaces, even allowing the definition of Wasserstein distances on Wasserstein spaces $\mathcal{P}_2(\mathcal{X})$ (Bonet et al., 2025b). This is particularly useful for comparing distributions over non-Euclidean objects. For example, Euclidean distances as ground metric between two images often yield poor results (Stanczuk et al., 2021), whereas Wasserstein distances are robust to small image perturbations (Beckmann et al., 2025). Similarly, comparing point clouds (Nguyen et al., 2021b) is natural with OT but not even well-defined with Euclidean distances. While most OT applications focus either on comparing pairwise objects or distributions over Euclidean spaces, recent work leverages Wasserstein distances on Wasserstein spaces for non-Euclidean domains, such as image (Dukler et al., 2019) or point cloud (Haviv et al., 2025) distributions.

The underlying concept of multilevel OT has been introduced under various names, including hierarchical OT (Schmitzer & Schnörr, 2013; Lee et al., 2019), mixture Wasserstein (Chen et al., 2018; 2019; Delon & Desolneux, 2020), and Wasserstein over Wasserstein (WoW) (Bonet et al., 2025b). It has applications beyond images and shapes, including domain adaptation (Lee et al., 2019; El Hamri et al., 2022), single-cell analysis (Lin et al., 2023), point cloud registration (Steuernagel et al., 2023), Bayesian inference (Nguyen & Mueller, 2024), generative modelling (Atanackovic et al., 2025; Haviv et al., 2025), document analysis (Yurochkin et al., 2019), Gromov–Wasserstein approximations (Mémoli, 2011; Piening & Beinert, 2025a), and reinforcement learning (Ziesche & Rozo, 2023). Extending this framework with another Polish space $\mathcal{Y}$ for dataset labels yields the OT dataset distance (OTDD), defined on $\mathcal{P}_2(\mathcal{Y} \times \mathcal{P}_2(\mathcal{X}))$ (Alvarez-Melis & Fusi, 2020). However, all these approaches incur high computational cost due to repeated pairwise Wasserstein evaluations.

Due to the complexity of OT, sliced Wasserstein distances (Bonneel et al., 2015; Nguyen, 2025) provide efficient OT-based alternatives to standard Wasserstein distances. Initially developed for probability measures on Euclidean spaces, they have since been extended to the sphere (Bonet et al., 2023a; Quellmalz et al., 2023), manifolds (Bonet et al., 2025a), functions (Garrett et al., 2024), hyperbolic spaces (Bonet et al., 2023b), the rotation group (Quellmalz et al., 2024), and matrices (Bonet et al., 2023c). Repeated sliced Wasserstein distances can be linearized by mapping measures onto function spaces (Naderializadeh et al., 2021; Nguyen, 2025). For WoW-type distances, sliced accelerations have been proposed for Gaussian mixtures (Nguyen & Mueller, 2024; Piening & Beinert, 2025b) and more generally for measures over measures (*meta-measures*) via sliced Wasserstein Busemann Gaussian (SWBDG/SWB1DG) distances (Bonet et al., 2025c) and the sliced OTDD (s-OTDD) (Nguyen et al., 2025). The SWBDG and SWB1DG, which appeared after the original submission of our work, are based on Busemann functions whose level sets act as a natural generalization of affine hyperplanes in the space of meta-measures. The numerical implementation relies on Gaussian approximations and a closed-form of the Busemann function. As an alternative, the s-OTDD employs a hierarchical slicing approach based on the method of moments. However, it is only well-defined for finite moments, and practical implementations are limited to the first few moments because of numerical instability – originally, the first five.

In this paper, we aim to circumvent the issues of the s-OTDD. Therefore, we build on theoretical ideas originally proposed in (Han, 2023) to develop a computable sliced Wasserstein metric on general Banach spaces. Employing the isometry between 1d probability measures in the Wasserstein space and quantile functions embedded in the space of square-integrable functions $L_2([0, 1])$, we utilize this to define a sliced Wasserstein metric on the space of 1d meta-measures via $L_2$-projections. Due to the lack of a uniform distribution on the unit ball of infinite-dimensional function spaces, we parametrize our projection directions as Gaussian processes (Kanagawa et al., 2018). To extend this idea to multi-dimensional meta-measures, we combine this approach with a classical slicing approach, mapping these meta-measures to 1d meta-measures. Lastly, we prove that the minimization of our sliced distance results in WoW minimization. This leads to the following contributions:

- We generalize the sliced Wasserstein distance to arbitrary Banach spaces. Moreover, we show how two distinct parameterizations of the random projections may result in equivalent metrics. As a special case, this allows for a sliced distance between 1d meta-measures.

- Beyond 1d meta-measures, we extend our approach to the multivariate case by introducing the double-sliced Wasserstein (DSW[1]) metric between meta-measures. Illustrating the usefulness of our DSW metric as a WoW replacement, we prove a form of topological metric equivalence between the two for discretized meta-measures.

- Lastly, we present various numerical experiments showcasing the advantages of our approach, allowing for meaningful distribution comparisons for datasets, shapes, and images.

## 2 WASSERSTEIN DISTANCES

The so-called Wasserstein distance or Kantorovich–Rubinstein metric is an optimal transport-based similarity gauge between probabilities on a common Polish space. To this end, let $\mathcal{X}$ be a Polish space, let $\mathcal{P}(\mathcal{X})$ be the space of Borel probability measures on $\mathcal{X}$ with respect to the Borel $\sigma$-algebra induced by the underlying metric, and let $\mathcal{P}_p(\mathcal{X}) := \{\mu \in \mathcal{P}(\mathcal{X}) \mid \exists x_1 \in \mathcal{X} : \int_{\mathcal{X}} d^p(x_1, x_2) \, d\mu(x_2) < \infty\}$, $p \in [1, \infty)$, be the subset of measures with finite $p$th moment. For $\mu \in \mathcal{P}(\mathcal{X})$ and a second Polish space $\mathcal{Y}$, the *push-forward* by a measurable map $T : \mathcal{X} \to \mathcal{Y}$ is defined by $T_\sharp \mu := \mu \circ T^{-1} \in \mathcal{P}(\mathcal{Y})$. The set of *transport plans* between $\mu \in \mathcal{P}(\mathcal{X}), \nu \in \mathcal{P}(\mathcal{Y})$ is given by

$$\Gamma(\mu, \nu) := \{\gamma \in \mathcal{P}(\mathcal{X} \times \mathcal{Y}) \mid \pi_{1,\sharp} \gamma = \mu, \pi_{2,\sharp} \gamma = \nu\},$$

where $\pi_i$ denotes the canonical projection onto the $i$th component. For a complete, separable metric space $(\mathcal{X}, d)$, the *(2-)Wasserstein distance*

$$\mathrm{W}(\mu, \nu; \mathcal{X}) := \inf_{\gamma \in \Gamma(\mu, \nu)} \left( \int_{\mathcal{X} \times \mathcal{X}} d^2(x_1, x_2) \, d\gamma(x_1, x_2) \right)^{\frac{1}{2}}$$

---

[1]Note that the acronym DSW is also used for the distributional sliced Wasserstein distance (Nguyen et al., 2021a), which modifies the slicing distribution of the classical sliced Wasserstein distance.

defines a metric on $\mathcal{P}_2(\mathcal{X})$. More precisely, $(\mathcal{P}_2(\mathcal{X}), \mathrm{W})$ is again a complete separable metric space, allowing the construction of the so-called *Wasserstein over Wasserstein* (WoW) *distance* $\mathbf{W}(\cdot, \cdot; \mathcal{X}) := \mathrm{W}(\cdot, \cdot; \mathcal{P}_2(\mathcal{X}))$, which is studied in (Bonet et al., 2025b).

In difference to other similarity gauges like the Kullback–Leibler divergence or the total variation, Wasserstein distances leverage the underlying geometry, allowing for meaningful comparisons between empirical measures. Although the Wasserstein distance relies on a linear program, the actual calculation is computationally costly. For two empirical measures supported at $n$ points in $\mathbb{R}^d$ equipped with the Euclidean metric, the exact computation has complexity $\mathcal{O}(n^3 \log n)$. The approximate computation based on entropic regularization still has complexity $\mathcal{O}(n^2 \log n)$, see (Peyré & Cuturi, 2019). Notably, this computational burden becomes even more involved for non-Euclidean metric spaces, where the computation of the underlying distance itself is challenging. For instance, the computation of the WoW distance relies on the pointwise evaluations of Wasserstein distances. If the empirical meta-measures in $\mathcal{P}_2(\mathcal{P}_2(\mathbb{R}^d))$ are supported on $N$ empirical measures on $\mathcal{P}_2(\mathbb{R}^d)$, each with $n$ support points, then the approximate calculation of the required distance matrix already has complexity $\mathcal{O}(N^2 n^2 \log n)$.

From a computational point of view, the Wasserstein distance on $(\mathbb{R}, |\cdot - \cdot|)$ is a notable exception since this may be evaluated analytically. To this end, for $\mu \in \mathcal{P}(\mathbb{R})$, its *quantile function* $Q_\mu \colon (0,1) \to \mathbb{R}$ is given by $Q_\mu(s) := \inf \{ x \in \mathbb{R} \mid \mu((-\infty, x]) \geq s \}$. The Wasserstein distance now becomes

$$\mathrm{W}(\mu, \nu; \mathbb{R}) = \left( \int_0^1 |Q_\mu(s) - Q_\nu(s)|^2 \, \mathrm{d}s \right)^{\frac{1}{2}}, \quad \mu, \nu \in \mathcal{P}_2(\mathbb{R}),$$

meaning that the *quantile mapping*

$$q \colon \mathcal{P}_2(\mathbb{R}) \to L_2([0,1]), \quad \mu \mapsto Q_\mu \tag{1}$$

is an isometric embedding into the space of square-integrable functions, see (Villani, 2003). For empirical measures, the quantile functions are piecewise constant and can be efficiently computed by sorting the support points.

## 3    SLICED WASSERSTEIN DISTANCES

At their core, all sliced Wasserstein distances exploit easy-to-compute, 1d optimal transports to define efficient alternatives to the standard Wasserstein distance. Originally, the sliced Wasserstein distance has been studied for measures in $\mathcal{P}_2(\mathbb{R}^d)$ and is based on the *slicing operator*

$$\pi_\theta \colon \mathbb{R}^d \to \mathbb{R}, \ x \mapsto \langle \theta, x \rangle, \qquad \theta \in \mathbb{S}^{d-1} := \{ x \in \mathbb{R}^d \mid \|x\| = 1 \}, \tag{2}$$

with respect to the Euclidean inner product and norm. The *sliced (2-)Wasserstein distance* reads as

$$\mathrm{SW}(\mu, \nu) := \left( \int_{\mathbb{S}^{d-1}} \mathrm{W}^2(\pi_{\theta, \sharp} \mu, \pi_{\theta, \sharp} \nu; \mathbb{R}) \, \mathrm{d}\mathbb{S}^{d-1}(\theta) \right)^{\frac{1}{2}}, \tag{3}$$

where we integrate with respect to the uniform probability on $\mathbb{S}^{d-1}$. Similar to the Wasserstein distance, SW metricizes the weak convergence (Bonnotte, 2013; Nadjahi et al., 2020). The spherical integral is usually approximated by Monte Carlo schemes (Bonneel et al., 2015; Nguyen et al., 2024; Hertrich et al., 2025) or Gaussian approximation (Nadjahi et al., 2021).

### 3.1    SLICING INFINITE DIMENSIONAL BANACH SPACES

As preliminary step towards an implementable sliced WoW distance, we consider the slicing on an infinite dimensional, separable Banach space $U$ with norm $\|\cdot\|$. We denote the continuous dual by $U^*$ and equip it with the dual norm, i.e., $\|v\| := \sup\{|v(u)| \mid u \in U, \|u\| \leq 1\}$ for all $v \in U^*$. Relying on the dual pairing, we generalize the slicing operator (2) by

$$\pi_v \colon U \to \mathbb{R}, \ u \mapsto \langle v, u \rangle := v(u), \qquad v \in U^*.$$

The crucial point in defining a sliced distance on $\mathcal{P}_2(U) := \{ \mu \in \mathcal{P}(U) \mid \int_U \|u\|^2 \, \mathrm{d}\mu(x) < \infty \}$. is that there exists no uniform probability measure on the infinite dimensional sphere. As remedy, we

choose an arbitrary $\xi \in \mathcal{P}_2(U^*) := \{\zeta \in \mathcal{P}(U^*) \mid \int_{U^*} \|v\|^2 \, \mathrm{d}\zeta(v) < \infty\}$ and define the $\xi$-based *sliced Wasserstein distance* as

$$\mathrm{SW}(\mu, \nu; \xi) := \left( \int_{U^*} \mathrm{W}^2(\pi_{v,\sharp} \mu, \pi_{v,\sharp} \nu; \mathbb{R}) \, \mathrm{d}\xi(v) \right)^{\frac{1}{2}}, \quad \mu, \nu \in \mathcal{P}(U). \tag{4}$$

This approach extends the slicing on Hilbert spaces (Han, 2023). However, unlike (Han, 2023), we do not construct specific measures on the sphere. This allows the use of easy-to-sample slicing measures. If the support of $\xi$ covers all directions in $U^*$, the $\xi$-based SW distance becomes a metric. Here, the crucial point is the definiteness. For this, we show the Lipschitz continuity of the integrant in (4) and exploit the uniqueness of the characteristic function, see Appendix A.

**Theorem 3.1.** *For $\xi \in \mathcal{P}_2(U^*)$, the $\xi$-based SW distance defines a well-defined pseudo-metric. If* $\operatorname{supp} \xi \cap \operatorname{span} v \notin \{\emptyset, \{0\}\}$ *for all* $v \in U^* \setminus \{0\}$*, then* (4) *defines a metric on* $\mathcal{P}_2(U)$*.*

If the slicing measure $\xi$ has full support, then the assumption in Theorem 3.1 is fulfilled, and the $\xi$-based SW distance is a metric. Two measures $\xi_1, \xi_2 \in \mathcal{P}_2(U^*)$ are equivalent if they are mutually absolutely continuous. If their Radon–Nikodým derivatives $\mathrm{d}\xi_1 / \mathrm{d}\xi_2$ and $\mathrm{d}\xi_2 / \mathrm{d}\xi_1$ are bounded, then the resulting SW distances are metrically equivalent. The proof is given in Appendix A.

**Proposition 3.2.** *Let $\xi_1, \xi_2 \in \mathcal{P}_2(U^*)$ be equivalent. If $\mathrm{d}\xi_1 / \mathrm{d}\xi_2$ and $\mathrm{d}\xi_2 / \mathrm{d}\xi_1$ are bounded, then we find $c_1, c_2 > 0$ such that*

$$c_1 \, \mathrm{SW}(\mu, \nu; \xi_1) \le \mathrm{SW}(\mu, \nu; \xi_2) \le c_2 \, \mathrm{SW}(\mu, \nu; \xi_1) \quad \forall \mu, \nu \in \mathcal{P}_2(U).$$

In the finite-dimensional Euclidean setting, special cases of the $\xi$-based SW distance correspond, for instance, to so-called energy measures on the sphere (Nguyen & Ho, 2023) and the standard Gaussian (Nadjahi et al., 2021). Relying on the latter, we obtain a strong equivalence to original SW (3) if $\xi$ is equivalent to the standard Gaussian. The short proof, which relies on Proposition 3.2 and the fact that the classical SW distance (3) and the reference-based SW distance (4) with the standard Gaussian reference coincide on $\mathcal{P}_2(\mathbb{R}^d)$, is given in Appendix A.

**Proposition 3.3.** *Let $\xi \in \mathcal{P}_2(\mathbb{R}^d)$ be equivalent to $\eta \sim \mathcal{N}(0, \mathbf{I}_d)$. If $\mathrm{d}\xi / \mathrm{d}\eta$ and $\mathrm{d}\eta / \mathrm{d}\xi$ are bounded, then we find $c_1, c_2 > 0$ such that*

$$c_1 \, \mathrm{SW}(\mu, \nu; \xi) \le \mathrm{SW}(\mu, \nu; \eta) = \mathrm{SW}(\mu, \nu) \le c_2 \, \mathrm{SW}(\mu, \nu; \xi) \quad \forall \mu, \nu \in \mathcal{P}_2(\mathbb{R}^d).$$

Assuming that samples of $\xi$ are available, the $\xi$-based SW distance on every separable Banach space can again be computed using the Monte Carlo scheme. If $\xi$ has a finite fourth moment, i.e. $\xi \in \mathcal{P}_4(U^*)$, the Monte Carlo scheme converges. The details are given in Proposition A.6.

## 3.2 SLICING THE 1D WASSERSTEIN SPACE

Exploiting the generalized SW distance in (4), we introduce a first slicing of the Wasserstein space $(\mathcal{P}_2(\mathbb{R}), \mathrm{W})$, which later builds the foundation of our sliced WoW distance on $\mathcal{P}_2(\mathcal{P}_2(\mathbb{R}^d))$. Recall that the quantile mapping (1) is an isometric embedding and thus measurable. Therefore, we can push every meta-measure $\boldsymbol{\mu} \in \mathcal{P}_2(\mathcal{P}_2(\mathbb{R}))$ to $q_\sharp \boldsymbol{\mu} \in \mathcal{P}_2(L_2([0,1]))$. In this manner, the WoW distance between $\boldsymbol{\mu}, \boldsymbol{\nu} \in \mathcal{P}_2(\mathcal{P}_2(\mathbb{R}))$ becomes $\mathbf{W}(\boldsymbol{\mu}, \boldsymbol{\nu}; \mathbb{R}) = \mathrm{W}(q_\sharp \boldsymbol{\mu}, q_\sharp \boldsymbol{\nu}; \mathcal{P}_2(L_2([0,1])))$. Fixing $\xi \in \mathcal{P}_2(L_2([0,1]))$, we introduce the *sliced quantile WoW* (SQW) *distance:*

$$\mathbf{SQW}(\boldsymbol{\mu}, \boldsymbol{\nu}; \xi) := \mathrm{SW}(q_\sharp \boldsymbol{\mu}, q_\sharp \boldsymbol{\nu}; \xi), \qquad \boldsymbol{\mu}, \boldsymbol{\nu} \in \mathcal{P}_2(\mathcal{P}_2(\mathbb{R})). \tag{5}$$

If $\xi$ is *positive*, i.e., $\xi$ has full support, then the assumptions of Theorem 3.1 are satisfied.

**Corollary 3.3.1.** *Let $\xi \in \mathcal{P}_2(L_2([0,1]))$ be positive, then $\mathbf{SQW}$ is a metric on $\mathcal{P}_2(\mathcal{P}_2(\mathbb{R}))$.*

For the later implementation, we require an easy-to-sample slicing measure. To this end, we propose to use Gaussians, i.e., measures $\xi \in \mathcal{P}_2(L_2([0,1]))$ such that $\pi_{v,\sharp} \xi$ is Gaussian for all $v \in L_2([0,1])$, see (Bogachev, 1998). On $L_2([0,1])$, there exists a one-to-one correspondence between Gaussian measures and Gaussian processes (Rajput & Cambanis, 1972, Thm. 2). In our numerics, we restrict ourselves to the Gaussian process $G$ that is related to the covariance kernel

$$k_\sigma \colon [0,1] \times [0,1] \to \mathbb{R}, \quad (t,s) \mapsto \exp(-|t-s|^2 / 2\sigma^2). \tag{6}$$

This means that we consider the function-valued random variable $G$ with

$$\big(G(t_1), \dots, G(t_n)\big) \sim \mathcal{N}\big(0, (k(t_i, t_j))_{i,j=1,\dots,n}\big) \qquad \forall t_1, \dots, t_n \in [0,1].$$

Since the kernel is smooth, the sample paths (realizations) of $G$ are smooth too, i.e., $G \in \mathcal{C}^\infty([0,1])$ almost surely, see (Da Costa et al., 2026, Cor. 1). Since $k_\sigma$ is *universal* (Steinwart, 2001), the corresponding Gaussian measure has full support (Van Der Vaart et al., 2008).

## 4 DOUBLE-SLICING THE WASSERSTEIN SPACE

The slicing schemes in § 3.1 and § 3.2 cannot directly be generalized to the multidimensional Wasserstein space $(\mathcal{P}_2(\mathbb{R}^d), \mathrm{W})$ due to the lack of the Banach space structure and since we do not have an adequate generalization of the quantile mapping. Instead of slicing the Wasserstein space directly, in a first step, we therefore propose to slice the underlying domain using

$$\boldsymbol{\pi}_\theta \colon \mathcal{P}_2(\mathbb{R}^d) \to \mathcal{P}_2(\mathbb{R}), \quad \mu \mapsto \pi_{\theta,\sharp}\mu, \qquad \theta \in \mathbb{S}^{d-1}.$$

with $\pi_\theta$ from (2). Notice that $\boldsymbol{\pi}_\theta$ is continuous with respect to the Wasserstein distances and thus measurable. This allows us to define the *sliced WoW distance* via

$$\mathbf{SW}(\boldsymbol{\mu}, \boldsymbol{\nu}) := \left( \int_{\mathbb{S}^{d-1}} \mathbf{W}^2(\boldsymbol{\pi}_{\theta,\sharp}\boldsymbol{\mu}, \boldsymbol{\pi}_{\theta,\sharp}\boldsymbol{\nu}; \mathbb{R}) \, \mathrm{d}\mathbb{S}^{d-1}(\theta) \right)^{\frac{1}{2}}, \qquad \boldsymbol{\mu}, \boldsymbol{\nu} \in \mathcal{P}_2(\mathcal{P}_2(\mathbb{R}^d)),$$

where we again integrate with respect to the uniform measure. Based on finite-dimensional slicing, $\mathbf{SW}$ essentially reduces meta-measures $\boldsymbol{\mu}$ and $\boldsymbol{\nu}$ to a series of 1d meta-measures $\boldsymbol{\pi}_{\theta,\sharp}\boldsymbol{\mu}$ and $\boldsymbol{\pi}_{\theta,\sharp}\boldsymbol{\nu}$.

As the computation of the 1d WoW distance remains challenging, we resort to a hierarchical slicing approach, inspired by (Nguyen et al., 2025; Piening & Beinert, 2025b). Using quantile slicing (5) with slicing measure $\xi \in \mathcal{P}_2(L_2([0,1]))$, we introduce the *double-sliced WoW distance*

$$\mathbf{DSW}(\boldsymbol{\mu}, \boldsymbol{\nu}; \xi) := \left( \int_{\mathbb{S}^{d-1}} \mathbf{SQW}^2(\boldsymbol{\pi}_{\theta,\sharp}\boldsymbol{\mu}, \boldsymbol{\pi}_{\theta,\sharp}\boldsymbol{\nu}; \xi) \, \mathrm{d}\mathbb{S}^{d-1}(\theta) \right)^{\frac{1}{2}}, \qquad \boldsymbol{\mu}, \boldsymbol{\nu} \in \mathcal{P}_2(\mathcal{P}_2(\mathbb{R}^d)).$$

By construction, the double-sliced WoW distance defines at least a pseudo-metric. If we restrict ourselves to empirical meta-measure, $\mathbf{DSW}$ even becomes a metric that is weakly equivalent to $\mathbf{W}$. To be more precise, we denote the subset of empirical measures by $\mathcal{P}_\mathrm{e}$ and the Dirac measure by $\delta_\bullet$. An *empirical meta-measure* $\boldsymbol{\mu} \in \mathcal{P}_\mathrm{e}(\mathcal{P}_\mathrm{e}(\mathbb{R}^d))$ has the form

$$\boldsymbol{\mu} = \frac{1}{N} \sum_{i=1}^{N} \delta_{\mu_i} \quad \text{with} \quad \mu_i = \frac{1}{n_i} \sum_{k=1}^{n_i} \delta_{x_{i,k}} \quad \text{and} \quad x_{i,k} \in \mathbb{R}^d \tag{7}$$

for arbitrary $N$ and $n_i$. For $N$ and $n_i \equiv \tilde{n}$ fixed, we denote $\boldsymbol{\mu} \in \mathcal{P}_\mathrm{e}^N(\mathcal{P}_\mathrm{e}^{\tilde{n}}(\mathbb{R}^d))$ with $\mu_i \in \mathcal{P}_\mathrm{e}^{\tilde{n}}(\mathbb{R}^d)$.

**Theorem 4.1.** *For positive $\xi \in \mathcal{P}_2(L_2([0,1]))$, $\mathbf{DSW}$ defines a metric on $\mathcal{P}_\mathrm{e}(\mathcal{P}_\mathrm{e}(\mathbb{R}^d))$. Moreover, for $\boldsymbol{\mu}_n, \boldsymbol{\mu} \in \mathcal{P}_\mathrm{e}^N(\mathcal{P}_\mathrm{e}^{\tilde{n}}(\mathcal{X}))$ with compact $\mathcal{X} \subset \mathbb{R}^d$ and positive Gaussian $\xi$, it holds*

$$\mathbf{DSW}(\boldsymbol{\mu}_n, \boldsymbol{\mu}; \xi) \to 0 \iff \mathbf{SW}(\boldsymbol{\mu}_n, \boldsymbol{\mu}; \xi) \to 0 \iff \mathbf{W}(\boldsymbol{\mu}_n, \boldsymbol{\mu}; \mathbb{R}^d) \to 0 \quad as \quad n \to \infty.$$

The detailed proofs are given in Appendix B. There, a transport plan is constructed to rely on the metric properties of WoW. The equivalence is then proven via a discretization of the Gaussian process and the compactness of $\mathcal{P}_\mathrm{e}^N(\mathcal{P}_\mathrm{e}^{\tilde{n}}(\mathcal{X}))$. Numerically, $\mathbf{DSW}$ can be implemented combining several integration techniques, see Appendix C.1 and C.2. Here, we consider the Gaussian $\xi$ related to the kernel in (6) and empirical meta-measures $\boldsymbol{\mu}, \boldsymbol{\nu} \in \mathcal{P}_\mathrm{e}(\mathcal{P}_\mathrm{e}(\mathbb{R}^d))$ as in (7). To approximate the outer integral over $\mathbb{S}^{d-1}$ and the inner integral over $\xi$ simultaneously, we employ the Monte Carlo method. For a sample path $g$ of the corresponding Gaussian process and a direction $\theta \in \mathbb{S}^{d-1}$, we evaluate the integrand as follows: The first slicing gives $\boldsymbol{\pi}_{\theta,\sharp}\boldsymbol{\mu} = (1/N) \sum_{i=1}^{N} \delta_{\pi_{\theta,\sharp}\mu_i}$, and the quantile mapping $q_\sharp \boldsymbol{\pi}_{\theta,\sharp}\boldsymbol{\mu} = (1/N) \sum_{i=1}^{N} \delta_{q(\pi_{\theta,\sharp}\mu_i)}$, where the piecewise constant quantile functions can be determined by sorting the support points. For the slicing on $L_2([0,1])$, we employ a quadrature to approximate the inner product. To this end, for knots $t_1, \ldots, t_R \in [0,1]$ and weights $w_1, \ldots, w_R$, we estimate

$$\widehat{\pi_{g,\sharp}q_\sharp\boldsymbol{\pi}_{\theta,\sharp}\boldsymbol{\mu}} = \frac{1}{N} \sum_{i=1}^{N} \delta_{\widehat{\langle q(\pi_{\theta,\sharp}\mu_i), g \rangle}} \quad \text{with} \quad \widehat{\langle q(\pi_{\theta,\sharp}\boldsymbol{\mu}_i), g \rangle} = \sum_{r=1}^{R} w_r \, q(\pi_{\theta,\sharp}\mu_i)(t_r) \, g(t_r).$$

Note that the samples $g$ of the process $G$ satisfy $(G(t_1), \ldots, G(t_R)) \sim \mathcal{N}(0, k(t_r, t_{r'}))_{r,r'=1}^{R}$ and can be easily generated. Finally, the double-sliced WoW distance is computed by

$$\widehat{\mathbf{DSW}}(\boldsymbol{\mu}, \boldsymbol{\nu}) := \left( \frac{1}{S} \sum_{s=1}^{S} \mathrm{W}^2 \left( \widehat{\pi_{g_s,\sharp}q_\sharp\boldsymbol{\pi}_{\theta_s,\sharp}\boldsymbol{\mu}}, \widehat{\pi_{g_s,\sharp}q_\sharp\boldsymbol{\pi}_{\theta_s,\sharp}\boldsymbol{\nu}}; \mathbb{R} \right) \right)^{\frac{1}{2}}.$$

The remaining 1d Wasserstein distances can again be efficiently computed.

At its core, **DSW** requires the computation of the domain projection $\pi_\theta$ and a quantile projection $q$. As an alternative approach, s-OTDD considers a variation of $\pi_\theta$ with general 1d feature projections, including convolutions, and a (finite) moment projection instead of quantiles (Nguyen et al., 2025). Another concurrent work constructs a direct slicing of $\mathcal{P}_2(\mathcal{P}_2(\mathbb{R}^d))$ via so-called Busemann projections onto geodesic rays and relies on Gaussian approximations (Bonet et al., 2025c). Lastly, notice that the collection of these push-forward $q_\sharp \pi_{\theta,\sharp}$ can be interpreted as meta-measure version of the so-called sliced-Wasserstein embedding (Kolouri et al., 2016; Naderializadeh et al., 2021; Nguyen, 2025) that maps a classical measure in $\mathcal{P}_2(\mathbb{R}^d)$ to the quantile functions of its slices.

## 5 NUMERICAL EXPERIMENTS

In this section, we aim to showcase the numerical properties and benefits of our sliced distances. We start with the 1d case. Drawing a connection between meta-measures and the so-called Gromov–Wasserstein (GW) distance, we consider a shape classification experiment from (Piening & Beinert, 2025a). Next, we compare our multidimensional sliced distance to the s-OTDD (Nguyen et al., 2025). Finally, we present applications from the evaluation of point cloud distributions and perceptual image analysis. For all these experiments, we employ trapezoidal integration weights $w_r$. We refer to Appendix C for further experiments and details[2].

### 5.1 SHAPE CLASSIFICATION VIA LOCAL DISTANCE DISTRIBUTIONS

First, we repeat a shape classification experiment from (Piening & Beinert, 2025a) based on parametrizing shapes as so-called metric measure (mm-)spaces. A mm-space is a tuple $(\mathcal{X}, d_\mathcal{X}, \mu)$ consisting of a compact metric space $(\mathcal{X}, d_\mathcal{X})$ and a measure $\mu \in \mathcal{P}_2(\mathcal{X})$. Modelling data as (finite) metric spaces allows for invariance to isometric transformations such as rotations, often desirable for shapes. Particularly, we may parametrize 2d shapes as point clouds with pairwise Euclidean distances and 3d shapes as triangular meshes with pairwise surface distances (Beier et al., 2022).

While Gromov–Wasserstein (GW) distances define a metric between mm–spaces (Mémoli, 2011), computation is costly and relies on inexact non-convex minimization. As a remedy, alternatives employ pseudo-metrics via *local distance distributions*. Namely, they map a finite, uniformly-weighted mm-space $\mathbb{X} = (x_1, \ldots, x_N)$ with $\mu \in \mathcal{P}_e(\mathcal{X})$ to $\mathcal{P}_e(\mathcal{P}_e(\mathbb{R}))$ via the (non-injective) mapping

$$\mathbb{X} \mapsto \mu_\mathbb{X} := \frac{1}{N} \sum_{i=1}^{N} \delta_{d_\mathcal{X}(x_i,\cdot)_\sharp \mu}, \quad \delta_{d_\mathcal{X}(x_i,\cdot)_\sharp \mu} = \frac{1}{N} \sum_{j=1}^{N} \delta_{d_\mathcal{X}(x_i, x_j)}.$$

Now, we can represent two mm-spaces as $\mu_\mathbb{X}, \nu_\mathbb{Y} \in \mathcal{P}_e(\mathcal{P}_e(\mathbb{R}))$ and compare them using the 1d WoW distance (Mémoli, 2011, 'Third Lower Bound' (TLB')), another SW-based distance (Piening & Beinert, 2025a, 'Sliced Third Lower Bound' (STLB)) or a so-called energy distance (Sato et al., 2020, 'Anchor Energy' (AE)). Alternatively, we may use our SQW distance.

Repeating experiments from (Piening & Beinert, 2025a), we precompute pairwise distance matrices with respect to our SQW distance, TLB, STLB, AE, and the GW distance. Then, we estimate the accuracy of a k-nearest neighbor (KNN) classification by assigning each test point to the majority class among its three nearest neighbors. We average the classification accuracy over 1000 random 25%/75% training/test splits on four datasets. Based on preprocessing from (Piening & Beinert, 2025a), we employ the '2D Shapes' dataset (Beier et al. (2022), $N = 50$), the 'Animals' dataset (Sumner & Popović (2004), $N = 50$) and the 'FAUST' dataset (Bogo et al., 2014) with 500 ('FAUST-500') and 1000 ('FAUST-1000') vertices per shape. Additionally, we report results on a synthetic version of MNIST (LeCun et al., 1998) denoted 'MNIST-2000' ($N = 100$, 5 classes, 2000 points), whose details are given in Appendix C.4.1. We set the kernel parameter $\sigma$ to 0.01, $R = 10$ and $S = 100$. We use the same integration grid and projection number for STLB. We display the results in Table 1, where we additionally report the mean runtime of a single distance computation. We observe comparable performance across all distances and a runtime advantage of SQW and STLB for the large-scale FAUST-1000 and MNIST-2000 dataset, in particular against the GW distance. See Appendix C.4.2 for further studies.

---

[2]Code: `https://github.com/MoePien/slicing_wasserstein_over_wasserstein`

| Distance | 2D shapes | | Animals | | FAUST-500 | | FAUST-1000 | | MNIST-2000 | |
|---|---|---|---|---|---|---|---|---|---|---|
| | Acc. (%) | Time (ms) | Acc. (%) | Time (ms) | Acc. (%) | Time (ms) | Acc. (%) | Time (ms) | Acc. (%) | Time (ms) |
| **SQW (Ours)** | 99.5±1.2 | 0.9±0.3 | 99.1±1.3 | 0.9±0.1 | 38.6±5.7 | 9.8±5.7 | 42.7±5.9 | 13.8±15.1 | 84.8±4.7 | 48.6±2.8 |
| TLB | 100.0±0.3 | 0.5±0.4 | 100.0±0.0 | 0.4±0.1 | 36.7±5.6 | 27.2±15.8 | 40.2±6.0 | 60.1±9.6 | 88.7±4.5 | 298.7±25.8 |
| STLB | 99.5±1.2 | 0.8±1.0 | 99.3±1.8 | 0.8±0.1 | 37.6±5.6 | 9.8±5.7 | 39.4±5.6 | 14.0±14.9 | 84.1±5.0 | 45.1±2.8 |
| AE | 99.7±0.9 | 0.4±0.0 | 97.8±1.8 | 0.4±0.1 | 37.7±5.6 | 8.3±2.1 | 41.8±5.3 | 25.2±12.0 | 88.1±4.5 | 157.2±16.6 |
| GW | 99.7±0.6 | 1.3±5.7 | 100.0±0.0 | 2.5±1.1 | 29.2±4.4 | 266.4±94.7 | 33.0±5.3 | 1048.2±357.3 | —timeout— | |

Table 1: Shape classification with KNN: Mean accuracy (Acc., ↑) and runtime (Time).

| $S$ | $R$ | $\sigma$ | Acc (%) |
|---|---|---|---|
| $10^2$ | $10^1$ | $0.1^2$ | 84.8 |
| $10^3$ | $10^1$ | $0.1^2$ | 84.7 |
| $10^3$ | $10^2$ | $0.1^2$ | 84.7 |
| $10^2$ | $10^2$ | $0.1^2$ | 84.8 |
| $10^2$ | $10^1$ | $0.1^3$ | 85.0 |
| $10^2$ | $10^1$ | $0.1^1$ | 82.6 |

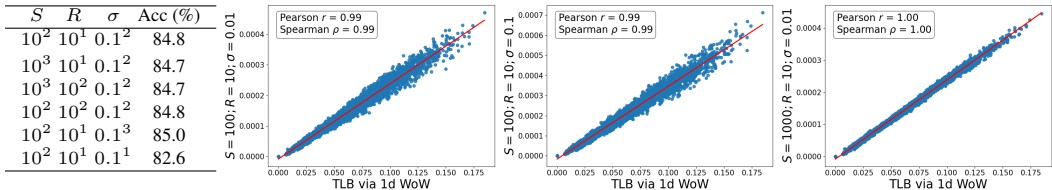

Figure 1: Left: Impact of projection number $S$, grid size $R$, and kernel parameter $\sigma$ on MNIST-2000 SQW classification. Right: Scatter plots and correlation between TLB (1d WoW, horizontal axis) and SQW (vertical axis) for different parameters on MNIST-2000.

Based on MNIST-2000, we additionally perform a parameter study to analyze the impact of the projection number $S$, grid size $R$, and kernel parameter $\sigma$. We again classify our MNIST-2000 dataset via SQW, but with varying parameters. The accuracies are displayed in the table in Figure 1, where we observe only slight variations of the resulting accuracies. Additionally, we analyze the correlation between TLB and SQW in Figure 1. The SQW distance is meant to emulate the WoW distance on $\mathcal{P}_2(\mathcal{P}_2(\mathbb{R}))$. Since TLB is exactly the WoW distance between $\mu_{\mathbb{X}}$ and $\mu_{\mathbb{Y}}$, we would therefore hope for a linear relation between SQW and TLB. To analyze this relation, we produce a scatter plot displaying all SQW-TLB pairs on MNIST-2000 for three different parameter configurations. In addition, we compute the Pearson and Spearman correlation coefficients. Overall, we see a clear linear relation with correlations coefficients of $0.99$ or more. Changing the kernel parameter $\sigma$ does not seem to impact this relation, but increasing the number of projections $S$ reduces the variance and raises the correlation from $0.99$ to $1.0$.

## 5.2 OPTIMAL TRANSPORT DATASET DISTANCE

Next, we consider a comparison with the OTDD (Alvarez-Melis & Fusi, 2020) and the s-OTDD (Nguyen et al., 2025). These metrics have been developed to quantify the similarities between labelled datasets in a model-agnostic manner. Such similarity metrics are especially important for applications in transfer learning. In this area, it has been shown empirically that the OTDD and s-OTDD display a strong correlation with the performance gap in transfer learning and classification accuracy in data augmentation, see (Alvarez-Melis & Fusi, 2020). To assess the suitability of our DSW distance as a drop-in replacement for the computationally costly OTDD, we repeat an experiment from (Nguyen et al., 2025). We randomly split MNIST (LeCun et al., 1998), FashionMNIST (Xiao et al., 2017), and CIFAR10 (Krizhevsky, 2009) to create subdataset pairs, each ranging in size from 500 to 1000, and compute the OTDD, the s-OTDD, and our DSW between subdataset pairs.

We plot the results of our DSW distance and the s-OTDD against OTDD for 100 dataset splits in Figure 2, where we include the Pearson and the Spearman correlation coefficients between both sliced distances and the OTDD. As our DSW distance is originally defined on $\mathcal{P}_2(\mathcal{P}_2(\mathbb{R}^d))$ and OTDD and s-OTDD on $\mathcal{P}(\mathcal{Y} \times \mathcal{P}(\mathbb{R}^d))$, we compute OTDD and s-OTDD with the label metric on $\mathcal{Y}$ set to zero for comparability, effectively representing each dataset as an empirical measure in $\mathcal{P}_2(\mathcal{P}_2(\mathbb{R}^d))$ and computing the OTDD via WoW. To be precise, each class-conditional distribution is modeled as $\mu_i \in \mathcal{P}_e(\mathbb{R}^d)$ and the distribution over class-conditional distributions becomes our meta-measure $\mu \in \mathcal{P}_e(\mathcal{P}_e(\mathbb{R}^d))$. Similar to (Nguyen et al., 2025), we estimate both DSW ($R = 10$, $\sigma = 0.1$) and the s-OTDD (with Radon features) with $S = 10,000$ projections. Based on this setting, we employ the original default implementation for the s-OTDD and the 'exact' OTDD.

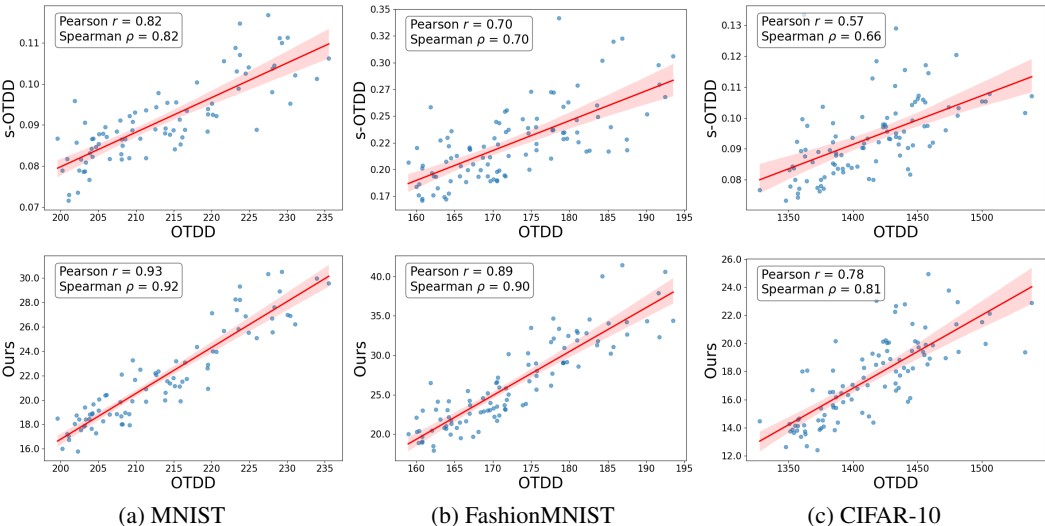

Figure 2: Scatter plots and correlations (↑) between the s-OTDD and the OTDD (top) and our DSW ('Ours') and the OTDD (bottom) for MNIST (2a), FashionMNIST (2b), and CIFAR-10 (2c).

Treating OTDD as our ground truth, we clearly see a stronger correlation between DSW and OTDD for all datasets. Additional experiments can be found in Appendix C.5.

## 5.3 COMPARING DISTRIBUTIONS OF POINT CLOUDS

For our next experiment, we assess the potential of DSW for evaluating point cloud generative models, which aim to generate 3D shapes such as chairs or planes. Evaluating such models is challenging since common quality metrics are insensitive to mode collapse (e.g., 'coverage') or tolerate low-quality samples (e.g., 'minimum matching distances'); see Yang et al. (2019) for details. A common remedy is the OT nearest-neighbor accuracy ('OT-NNA') test, which uses 1-nearest-neighbor classification based on pairwise Wasserstein distances between real and generated point clouds. However, for $N$ real and $M$ generated shapes, this requires $(N + M - 1)^2/2$ OT computations, without defining a true metric.

As a natural alternative, one might instead represent batches of real and generated point clouds $\mu_i \in \mathcal{P}_e(\mathbb{R}^3)$ as empirical meta-measures in $\mathcal{P}_e(\mathcal{P}_e(\mathbb{R}^3))$ and compute the WoW, respectively, the DSW distance between a real and a generated batch. To assess the suitability of our resulting quality metric, we consider shapes from ModelNet-10 (Wu et al., 2015) and construct meta-measures

$$\boldsymbol{\mu} = \frac{1}{N} \sum_{i=1}^{N} \delta_{\mu_i} \in \mathcal{P}_e(\mathcal{P}_e(\mathbb{R}^3)), \quad \boldsymbol{\nu} = \frac{1}{M} \sum_{j=1}^{M} \delta_{\nu_j} \in \mathcal{P}_e(\mathcal{P}_e(\mathbb{R}^3)),$$

where the support points are again Euclidean empirical measures

$$\mu_i = \frac{1}{n} \sum_{k=1}^{n} \delta_{x_{i,k}} \in \mathcal{P}_e(\mathbb{R}^3), \quad \nu_j = \frac{1}{m} \sum_{\ell=1}^{m} \delta_{x'_{j,\ell},+\varepsilon} \in \mathcal{P}_e(\mathbb{R}^3) \quad \epsilon \sim \mathcal{N}(0, \sigma_{\text{Noise}}^2 \mathbf{I}_3).$$

Given a certain shape class, e.g., 'chair', we initialize $\boldsymbol{\mu}$ as our fixed reference meta-measure with (downsampled) shapes from the ModelNet-10 training set and $\boldsymbol{\nu}$ as our varying target meta-measure with shapes from the ModelNet-10 test set. To compare OT-NNA, WoW, and our DSW metric, we independently vary the number of target shapes $M$, the level of Gaussian noise $\sigma_{\text{Noise}}$, and the point cloud discretization $m$ while fixing the remaining parameters according to the reference $\boldsymbol{\mu}$ (Default parameters: $N = M$, $\sigma_{\text{Noise}} = 0$, $m = n$). The results of our experiment are visualized in Figure 3, where we display the average result of 5 runs with varying $M$, $\sigma_{\text{Noise}}$, and $n$, and the reference ('ground truth') parameter of $\boldsymbol{\mu}$ is marked with a dotted red line. Lower is better for all metrics.

Looking at the number of target shapes $M$ in Subfigure 3a (Class: 'bed', $N = 10$, $\sigma_{\text{Noise}} = 0$, $n = m = 50$), we see that all metrics successfully capture mode collapse, i.e., $M = 1$, and decrease for a

larger number of target shapes $M$. Unlike the plateauing WoW and DSW metrics, OT-NNA displays an undesired behavior by increasing for $M \geq N = 10$, however. As for the random Gaussian perturbations in Subfigure 3b (Class: 'sofa', $N = M = 10$, $n = m = 50$), all three metrics increase with increasing noise. Whereas OT-NNA is more sensitive to small noise levels, the WoW and DSW metrics are more sensitive to high noise levels. Considering the point cloud resolution $m$ in Subfigure 3c (Class: 'monitor', $N = M = 10$, $n = 500$), OT-NNA seems inconsistent regarding the resolution. In contrast, the WoW and DSW metrics are higher for $m \leq 100$ and plateau after a seemingly sufficient resolution has been reached. Overall, we see that WoW and DSW display similar behavior as the OT-NNA and offer the advantage of being unbounded metrics. Additionally, DSW ($S = 10,000$, $R = 50$, $\sigma = 0.1$) offers computational advantages as it takes around 0.25 seconds for $M = N = 10$ and $m = n = 500$, where WoW takes about 4.5 seconds and OT-NNA about 8.5 seconds (on our CPU). This makes it especially suitable for high-resolution point clouds and large point cloud batches. Further studies are contained in Appendix C.6.

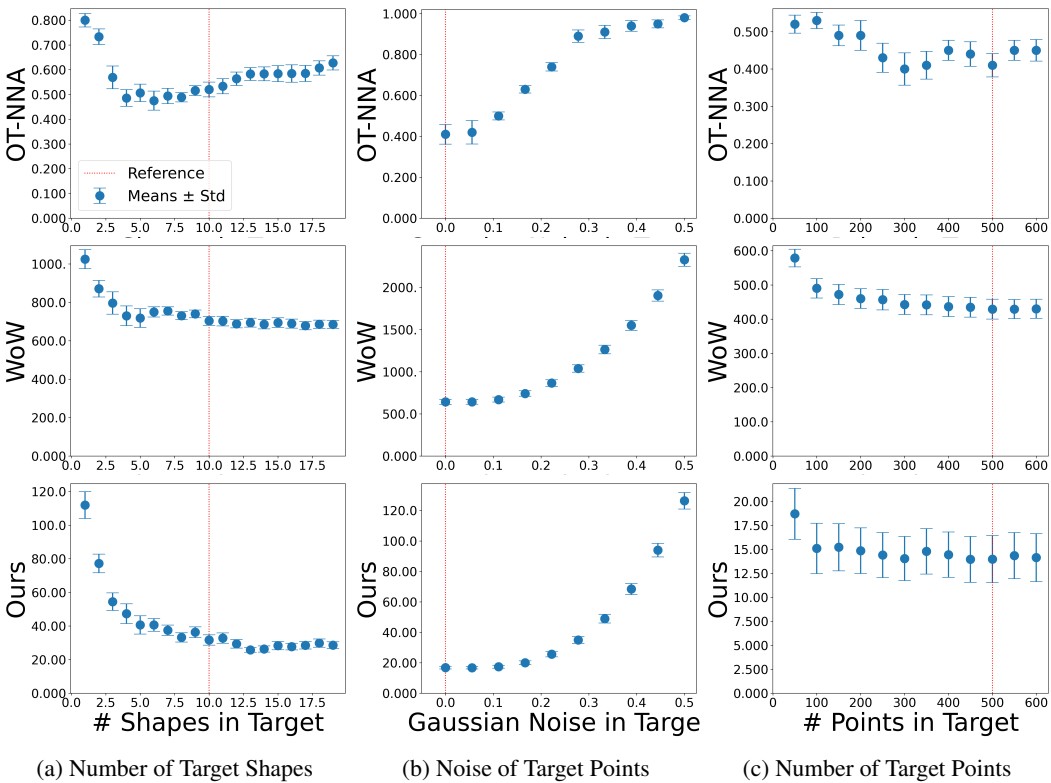

(a) Number of Target Shapes     (b) Noise of Target Points     (c) Number of Target Points

Figure 3: OT-NNA, WoW, and our DSW ('Ours')between target and reference point cloud batches for varying numbers of shapes $M$ in target batch (3a), for Gaussian noise $\sigma_{\text{Noise}}$ for target points (3b) and varying target point cloud resolution $m$ (3c). Fixed reference values are marked in red.

## 5.4 COMPARING IMAGE DISTRIBUTIONS VIA THEIR PATCH DISTRIBUTIONS

Given the importance of OT for imaging, we conclude with an imaging experiment. Interestingly, OT is utilized on two levels in this area. On the one hand, it is used to compare *two batches of images* using pairwise Euclidean distances (Genevay et al., 2018). On the other hand, using OT as a distance *between two individual images* remains of relevance due to the disadvantages of Euclidean distances. Those methods represent images as 2D histograms (Beier et al., 2023; Geuter et al., 2025) or as *patch* distributions (Hertrich et al., 2022; Elnekave & Weiss, 2022; Flotho et al., 2025). Thus, a natural combination is the comparison of image batches using WoW (Dukler et al., 2019).

Since patch-based OT distances serve as a perceptual metric between images (He et al., 2024), we incorporate patch-based image representations into the WoW framework. This is based on parametrizing images via their distribution of localized features. More concretely, we map each (grayscale)

image to the empirical measure over all contained (overlapping) square-shaped image regions of size $p \times p$, see (Piening et al., 2024) for an in-depth description. Thus, we may represent each image as an empirical measure $\mu_i \in \mathcal{P}_e(\mathbb{R}^{p^2})$ supported on vectorized patches and a batch of images as an empirical meta-measure $\boldsymbol{\mu} \in \mathcal{P}_e(\mathcal{P}_e(\mathbb{R}^{p^2}))$. In this experiment, we establish a quantitative comparison of image batches using our hierarchical OT framework as an alternative to comparing two image batches using standard OT or a neural-network-based perceptual metric, such as the 'Kernel Inception Distance' (Sutherland et al., 2018, 'KID').

To validate this approach, we consider synthetic $64 \times 64$ texture images based on random Perlin noise (Perlin, 1985). This texture synthesis model is controlled by several parameters, among them the *lacunarity* and the *persistence*, see Appendix C.7. Similar to our previous experiment, we initialize a reference meta-measure $\boldsymbol{\mu}$ over images represented as patch distributions according to some reference parameters and compare it to a target meta-measure $\boldsymbol{\nu}$ with varying lacunarity, respectively, persistence. For batch size 32 and patch size $p = 8$, both meta-measures are initialized according to 32 random images, where each image is represented by $(64 - 7) \times (64 - 7) = 3249$ uniformly weighted patches of dimension $8 \times 8 = 64$. In Figure 4, we compare the average behavior over five runs of the standard Wasserstein distance with Euclidean cost between our reference and target images and our patch-based DSW distance ($\sigma = 0.1$, $S = 10,000$ projections, $R = 10$), where the 'ground truth' lacunarity and persistence reference parameters are again marked in red.

We observe that both distances are minimized for the 'true' reference parameters. Still, our patch-based approach is more sensitive to parameter variation and better at discriminating between different batches, see also supplementary comparisons. Note that patch-based WoW calculation takes about 40 seconds, whereas our DSW distance merely requires about one second.

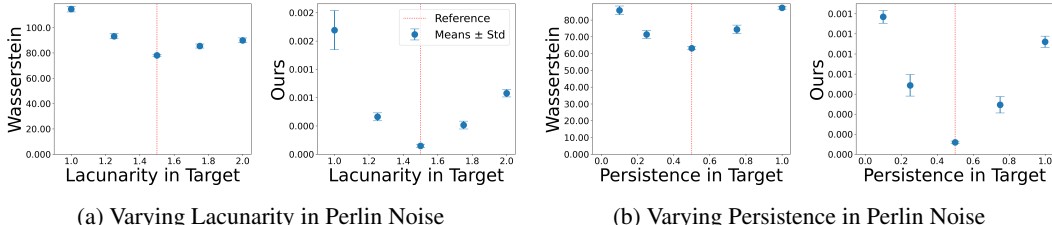

(a) Varying Lacunarity in Perlin Noise          (b) Varying Persistence in Perlin Noise

Figure 4: Comparing synthetic texture image batches via Euclidean Wasserstein and our sliced patch-based distance based on varying 'lacunarity' (4a) and 'persistence' (4b). Both distances are minimized for 'true' parameters (red), but our DSW ('Ours') distance leads to clearer discrimination.

## 6 CONCLUSION

We introduce a general sliced OT framework for measures on arbitrary Banach spaces. Leveraging the isometry between 1d Wasserstein and $L_2([0,1])$, Gaussian-process–parametrized $L_2$-projections, and classical spherical slicing, we define the DSW distance between meta-measures, a well-posed, computationally efficient substitute for WoW. We prove that DSW minimization corresponds to WoW minimization for discretized meta-measures and demonstrate practical effectiveness on datasets, shapes, and images.

On the practical side, future work could align DSW with the original OTDD by employing hybrid slicing (Nguyen & Ho, 2024) to extend DSW to $\mathcal{P}_2(\mathcal{Y} \times \mathcal{P}_2(\mathbb{R}^d))$ or integrate convolutional projections (Nguyen & Ho, 2022) similar to the s-OTDD. Also, one might employ our Banach slicing for infinite-dimensional generative models (Hagemann et al., 2025) or integrate dynamic transport into our framework to enable applications in generative flow matching similar to (Chapel et al., 2025).

On the theoretical side, it would be of interest to analyze further topological properties similar to Han (2023). Another interesting direction is the question of sample complexity: In high-dimensional Euclidean spaces, sliced Wasserstein distances require fewer samples to approximate a continuous probability measure than the Wasserstein distance (Nadjahi et al., 2020). As WoW distances require many samples to approximate a meta-measure (Catalano & Lavenant, 2024), it would be interesting to analyze if our DSW distance displays better sample complexity properties.

ACKNOWLEDGMENTS

MP gratefully acknowledges the financial support by the German Research Foundation (DFG), GRK2260 BIOQIC project 289347353. Moreover, we gratefully acknowledge fruitful discussions with Gabriele Steidl, Nicolaj Rux, and Gregor Kornhardt that helped improve the ideas in this work.

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

## A  Non-Spherical Sliced Wasserstein Distance on Banach Spaces

Here, we present proofs for Section 3.1. For clarity, we restate and prove our statements from the main paper as smaller statements.

**Lemma A.1.** *For $\mu, \nu \in \mathcal{P}_2(U)$, $\theta \in U^* \mapsto \mathrm{W}(\pi_{\theta,\sharp}\,\mu, \pi_{\theta,\sharp}\,\nu;\ \mathbb{R})$ is Lipschitz continuous.*

*Proof.* We generalize the proof of (Han, 2023, Lem. 2.3). For this, let $\theta_1, \theta_2 \in U^*$ be arbitrary. Using the triangle inequality and its reverse, we have

$$
\begin{aligned}
&|\mathrm{W}(\pi_{\theta_1,\sharp}\,\mu, \pi_{\theta_1,\sharp}\,\nu;\ \mathbb{R}) - \mathrm{W}(\pi_{\theta_2,\sharp}\,\mu, \pi_{\theta_2,\sharp}\,\nu;\ \mathbb{R})| \\
&\leq \big[\mathrm{W}(\pi_{\theta_1,\sharp}\,\mu, \pi_{\theta_2,\sharp}\,\mu;\ \mathbb{R}) + \mathrm{W}(\pi_{\theta_2,\sharp}\,\mu, \pi_{\theta_1,\sharp}\,\nu;\ \mathbb{R})\big] \\
&\qquad - \big[\mathrm{W}(\pi_{\theta_2,\sharp}\,\mu, \pi_{\theta_1,\sharp}\,\nu;\ \mathbb{R}) - \mathrm{W}(\pi_{\theta_1,\sharp}\,\nu, \pi_{\theta_2,\sharp}\,\nu;\ \mathbb{R})\big] \\
&= \mathrm{W}(\pi_{\theta_1,\sharp}\,\mu, \pi_{\theta_2,\sharp}\,\mu;\ \mathbb{R}) + \mathrm{W}(\pi_{\theta_1,\sharp}\,\nu, \pi_{\theta_2,\sharp}\,\nu;\ \mathbb{R}).
\end{aligned}
$$

For the first term on the left-hand side, it follows

$$
\begin{aligned}
\mathrm{W}^2(\pi_{\theta_1,\sharp}\,\mu, \pi_{\theta_2,\sharp}\,\mu;\ \mathbb{R}) &\leq \int_{\mathbb{R}^2} |t_1 - t_2|^2\,\mathrm{d}(\pi_{\theta_1}, \pi_{\theta_2})_\sharp\,\mu(t_1, t_2) = \int_U |\langle x, \theta_1 - \theta_2 \rangle|^2\,\mathrm{d}\mu(x) \\
&\leq \|\theta_1 - \theta_2\|_{U^*}^2 \int_U \|x\|_U^2\,\mathrm{d}\mu(x) = \|\theta_1 - \theta_2\|_{U^*}^2\, M_2(\mu),
\end{aligned}
$$

where $M_2(\mu) \coloneqq \int_U \|x\|_U^2\,\mathrm{d}\mu(x)$ is the second moment of $\mu$. Using an analogous estimate for the second term, we obtain

$$
|\mathrm{W}(\pi_{\theta_1,\sharp}\,\mu, \pi_{\theta_1,\sharp}\,\nu;\ \mathbb{R}) - \mathrm{W}(\pi_{\theta_2,\sharp}\,\mu, \pi_{\theta_2,\sharp}\,\nu;\ \mathbb{R})| \leq \|\theta_1 - \theta_2\|_{U^*}\,\big(M_2^{1/2}(\mu) + M_2^{1/2}(\nu)\big). \qquad \square
$$

This allows us to prove the first part of Theorem 3.1.

**Proposition A.2.** *For $\xi \in \mathcal{P}_2(U^*)$, the $\xi$-based SW distance is well-defined.*

*Proof.* The Lipschitz continuity in Lemma A.1 implies that the integrand in the formulation of the $\xi$-based SW (4) is measurable. To show that $\mathrm{SW}(\mu, \nu;\ \xi)$ is finite for $\mu, \nu \in \mathcal{P}_2(U)$, let $\gamma \in \Gamma(\mu, \nu)$ realize $\mathrm{W}(\mu, \nu;\ U)$. Because of

$$
\begin{aligned}
\mathrm{W}^2(\pi_{\theta,\sharp}\,\mu, \pi_{\theta,\sharp}\,\nu;\ \mathbb{R}) &\leq \int_{U \times U} |\langle x_1, \theta \rangle - \langle x_2, \theta \rangle|^2\,\mathrm{d}\gamma(x_1, x_2) \\
&\leq \|\theta\|_{U^*}^2 \int_{\mathcal{X} \times \mathcal{X}} \|x_1 - x_2\|_U^2\,\mathrm{d}\gamma(x_1, x_2) = \|\theta\|_{U^*}^2\,\mathrm{W}^2(\mu, \nu;\ U), \quad (8)
\end{aligned}
$$

the $\xi$-based SW distance is bounded by $\mathrm{SW}(\mu, \nu;\ \xi) \leq \mathrm{W}(\mu, \nu;\ U)\,M_2^{1/2}(\xi)$. $\qquad \square$

Now, we prove the second part of Theorem 3.1.

**Theorem A.3.** *Let $\xi \in \mathcal{P}_2(U^*)$ be such that $\mathrm{supp}\,\xi \cap \mathrm{span}\,\theta \notin \{\emptyset, \{0\}\}$ for any $\theta \in U^*$, then $\mathrm{SW}(\cdot, \cdot;\ \xi)$ defines a metric on $\mathcal{P}_2(U)$. Otherwise, $\mathrm{SW}(\cdot, \cdot;\ \xi)$ defines at least a pseudo-metric.*

*Proof.* Positivity, symmetry, and triangle inequality follow from the corresponding properties of the Wasserstein distance. For the definiteness, assume that $\mathrm{SW}(\mu, \nu;\ \xi) = 0$, which implies $\mathrm{W}(\pi_{\theta,\sharp}\,\mu, \pi_{\theta,\sharp}\,\nu;\ \mathbb{R}) = 0$ for almost all $\theta \in U^*$ with respect to $\xi$. The Lipschitz continuity in Lemma A.1 implies $\mathrm{W}(\pi_{\theta,\sharp}\,\mu, \pi_{\theta,\sharp}\,\nu;\ \mathbb{R}) = 0$ and thus $\pi_{\theta,\sharp}\,\mu = \pi_{\theta,\sharp}\,\nu$ for all $\theta \in \mathrm{supp}\,\xi$. Now, let $\theta' \in U^*$ be arbitrary. By assumption, we find $t \in \mathbb{R} \setminus \{0\}$ such that $t\theta' \in \mathrm{supp}\,\xi$. Furthermore, we have

$$
\begin{aligned}
\int_U \mathrm{e}^{\mathrm{i}\langle x, \theta' \rangle}\,\mathrm{d}\mu(x) &= \int_U \mathrm{e}^{\mathrm{i}\langle x, t\theta' \rangle \frac{1}{t}}\,\mathrm{d}\mu(x) = \int_{\mathbb{R}} \mathrm{e}^{\mathrm{i}s\frac{1}{t}}\,\mathrm{d}\pi_{t\theta',\sharp}\,\mu(s) \\
&= \int_{\mathbb{R}} \mathrm{e}^{\mathrm{i}s\frac{1}{t}}\,\mathrm{d}\pi_{t\theta',\sharp}\,\nu(s) = \int_U \mathrm{e}^{\mathrm{i}\langle x, \theta' \rangle}\,\mathrm{d}\nu(x).
\end{aligned}
$$

Since every measure on $U$ has a unique characteristic function, see (Ledoux & Talagrand, 1991, § 2.1), we conclude $\mu = \nu$. $\qquad \square$

We continue with Proposition 3.2.

**Proposition A.4.** *Let $\xi_1, \xi_2 \in \mathcal{P}_2(U^*)$ be equivalent. If $\mathrm{d}\xi_1/\mathrm{d}\xi_2$ and $\mathrm{d}\xi_2/\mathrm{d}\xi_1$ are bounded, then we find $c_1, c_2 > 0$ such that*

$$c_1 \operatorname{SW}(\mu, \nu; \xi_1) \leq \operatorname{SW}(\mu, \nu; \xi_2) \leq c_2 \operatorname{SW}(\mu, \nu; \xi_1) \quad \forall \mu, \nu \in \mathcal{P}_2(U).$$

*Proof.* Exploiting the bounded Radon–Nikodým derivatives, we obtain

$$\operatorname{SW}^2(\mu, \nu; \xi_2) = \int_{U^*} \operatorname{W}^2(\pi_{\theta,\sharp}\,\mu, \pi_{\theta,\sharp}\,\nu;\, \mathbb{R}) \tfrac{\mathrm{d}\xi_2}{\mathrm{d}\xi_1}(\theta)\,\mathrm{d}\xi_1(\theta) \leq \big\| \tfrac{\mathrm{d}\xi_2}{\mathrm{d}\xi_1}(\theta) \big\|_{L^\infty_{\xi_1}(U^*)} \operatorname{SW}^2(\mu, \nu; \xi_1).$$
$$\square$$

We employ this to prove Proposition 3.3.

**Proposition A.5.** *For $\xi \in \mathcal{P}(\mathbb{R}^d)$ absolutely continuous and $\mathrm{d}\xi/\mathrm{d}\eta$, $\mathrm{d}\eta/\mathrm{d}\xi$ bounded, for $\eta \sim \mathcal{N}(0, \mathbf{I}_d)$, there exist $c_1, c_2 > 0$ such that*

$$c_1 \operatorname{SW}(\mu, \nu\,;\, \xi) \leq \operatorname{SW}(\mu, \nu; \eta) = \operatorname{SW}(\mu, \nu) \leq c_2 \operatorname{SW}(\mu, \nu\,;\, \xi) \quad \forall \mu, \nu \in \mathcal{P}_2(\mathbb{R}^d).$$

*Proof.* This follows directly from Proposition A.4 and the identity $\operatorname{SW}(\mu, \nu) = \operatorname{SW}(\mu, \nu; \eta)$, see (Nadjahi et al., 2021, Prop. 1). $\square$

We extend our findings from the main paper with a statement about the computational approximation via Monte Carlo. In particular, the $\xi$-based SW distance can be numerically approximated by

$$\widehat{\operatorname{SW}}^2(\mu, \nu; \xi) \approx \frac{1}{S} \sum_{s=1}^{S} \operatorname{W}^2(\pi_{\theta_s,\sharp}\,\mu, \pi_{\theta_s,\sharp}\,\nu;\, \mathbb{R}), \quad \theta_s \sim \xi \text{ iid.}$$

It is well-known that such Monte Carlo estimates have a convergence rate of $\mathcal{O}(1/\sqrt{S})$ for $S$ random projections (Nadjahi et al., 2020). Given suitable conditions, a similiar results holds for this estimate.

**Proposition A.6.** *For $\xi \in \mathcal{P}_4(U^*)$, it holds*

$$\mathbb{E}_{\theta_1,\dots,\theta_S}|\widehat{\operatorname{SW}}^2(\mu, \nu;\, \xi) - \operatorname{SW}^2(\mu, \nu;\, \xi)| \leq \frac{1}{\sqrt{S}} \operatorname{std}_\theta \operatorname{W}^2(\pi_{\theta,\sharp}\,\mu, \pi_{\theta,\sharp}\,\nu;\, \mathbb{R}).$$

*Proof.* Using Hölder's inequality, and exploiting that the directions $\theta_s$ are independent and identically distributed, we have

$$\mathbb{E}_{\theta_1,\dots,\theta_S}|\widehat{\operatorname{SW}}^2(\mu, \nu;\, \xi) - \operatorname{SW}^2(\mu, \nu;\, \xi)|$$
$$\leq \Big( \int_{U^*} \cdots \int_{U^*} \Big| \frac{1}{S} \sum_{s=1}^{S} \operatorname{W}^2(\pi_{\theta_s,\sharp}\,\mu, \pi_{\theta_s,\sharp}\,\nu;\, \mathbb{R}) - \operatorname{SW}^2(\mu, \nu;\, \xi) \Big|^2 \,\mathrm{d}\xi(\theta_1) \cdots \mathrm{d}\xi(\theta_S) \Big)^{\frac{1}{2}}$$
$$= \frac{1}{\sqrt{S}} \Big( \sum_{s=1}^{S} \int_{U^*} |\operatorname{W}^2(\pi_{\theta_s,\sharp}\,\mu, \pi_{\theta_s,\sharp}\,\mu;\, \mathbb{R}) - \operatorname{SW}^2(\mu, \nu;\, \xi)|^2 \,\mathrm{d}\xi(\theta_s) \Big)^{\frac{1}{2}}$$
$$= \frac{1}{\sqrt{S}} \operatorname{std}_\theta \operatorname{W}^2(\pi_{\theta,\sharp}\,\mu, \pi_{\theta,\sharp}\,\nu;\, \mathbb{R}),$$

where the standard deviation exists due to $\operatorname{W}^4(\pi_{\theta,\sharp}\,\mu, \pi_{\theta,\sharp}\,\nu;\, \mathbb{R}) \leq \|\theta\|^4_{U^*} \operatorname{W}^4(\mu, \nu; U)$, cf. (8). $\square$

# B  DOUBLE-SLICED WASSERSTEIN DISTANCE

## B.1  METRIC PROPERTIES

To prove the positive definiteness of our double-sliced metric for empirical meta-measures, we utilize an extension of the 'Cramer–Wold' theorem by Cuesta-Albertos et al. (2007) about the set of projections required to separate measures on $\mathbb{R}^d$. The statement is based on the so-called *Carleman condition*. A measure $\mu \in \mathcal{P}_2(\mathbb{R}^d)$ fulfills this condition if all moments

$$M_p(\mu) := \int \|x\|^p \, \mathrm{d}\mu, \quad p \geq 1,$$

are finite and it holds that

$$\sum_{p=1}^{\infty} M_p^{-1/p} = \infty.$$

This condition is fulfilled for compactly supported measures (Schmüdgen et al., 2017, Ch. 14) and, in particular, empirical measures. We also refer to (Heppes, 1956; Tanguy et al., 2024) for similar results targeted at empirical measures.

**Lemma B.1.** *(Cuesta-Albertos et al., 2007, Corr. 3.2) Given measures $\mu, \nu \in \mathcal{P}_2(\mathbb{R}^d)$ that fulfill the Carleman condition and a set $S \subset \mathbb{S}^{d-1}$ of positive surface measure with*

$$\mathrm{W}(\pi_{\theta\sharp}\mu, \pi_{\theta\sharp}\nu; \mathbb{R}) = 0 \quad \text{for all } \theta \in S, \tag{9}$$

*it holds that $\mu = \nu$.*

This allows us to prove the metric properties presented in Theorem 3.1.

**Proposition B.2.** *Given a positive $\xi \in \mathcal{P}_2(L_2(Y))$, $\mathbf{DSW}$ defines a metric on $\mathcal{P}_e(\mathcal{P}_e(\mathbb{R}^d))$.*

*Proof.* 'Pseudo-metric': The symmetry and triangle inequality are trivial and follow directly from the ambient spaces of the embedded measures and the properties of the Wasserstein distance.

'Positive Definiteness': We aim to prove that

$$\mathbf{DSW}(\boldsymbol{\mu}, \boldsymbol{\nu}; \xi) = 0 \iff \boldsymbol{\mu} = \boldsymbol{\nu}.$$

for empirical meta-measures. Therefore, assume that $\mathbf{DSW}(\boldsymbol{\mu}, \boldsymbol{\nu}; \xi) = 0$ for $\boldsymbol{\mu}, \boldsymbol{\nu} \in \mathcal{P}_e(\mathcal{P}_e(\mathbb{R}^d))$.

Due to $\mathbf{DSW}(\boldsymbol{\mu}, \boldsymbol{\nu}; \xi) = 0$, we know that $\mathbf{SQW}(\boldsymbol{\pi}_{\theta,\sharp}\boldsymbol{\mu}, \boldsymbol{\pi}_{\theta,\sharp}\boldsymbol{\nu}; \xi) = 0$ for all $\theta \in \mathbb{S}^{d-1}$ except for a zero measure set $Z$. Since $\mathbf{SQW}$ is a metric on $\mathcal{P}_2(\mathbb{R})$, we thus know that $\boldsymbol{\pi}_{\theta,\sharp}\boldsymbol{\mu} = \boldsymbol{\pi}_{\theta,\sharp}\boldsymbol{\nu}$ for every $\theta \in \mathbb{S}^{d-1} \setminus Z$. Now, this means that there exists a $\boldsymbol{\gamma}_\theta^* \in \Gamma(\boldsymbol{\pi}_{\theta,\sharp}\boldsymbol{\mu}, \boldsymbol{\pi}_{\theta,\sharp}\boldsymbol{\nu}) \subset \mathbb{R}_{\geq 0}^{n \times m}$, such that

$$\langle \boldsymbol{\gamma}_\theta^*, C_\theta \rangle = 0, \tag{10}$$

where $C_\theta = (\mathrm{W}^2(\pi_{\theta,\sharp}\mu_i, \pi_{\theta,\sharp}\nu_j))_{i,j} \subset \mathbb{R}_{\geq 0}^{n \times m}$ and $\langle \cdot, \cdot \rangle$ denotes the Frobenius inner product. Here, our costs and transport plans take matrix form due to the empirical measure structure.

We now want to find a suitable transport plan from this set of transport plans to construct an upper bound for the meta-measure metric $\mathbf{W}(\boldsymbol{\mu}, \boldsymbol{\nu}; \mathbb{R}^d)$. Consider the set of index pairs $\mathrm{Idx} = \{(i,j) \mid \mu_i \neq \nu_j\}$. If this set is empty, then we are done. Otherwise, we know from Lemma B.1 that there exists no set $S \subset \mathbb{S}^{d-1}$ of positive measure for $(\mu_i, \nu_j)$, $(i,j) \in \mathrm{Idx}$, such that (9) is fulfilled for all $\theta \in S$. Conversely, for $(i,j) \in \mathrm{Idx}$, it holds that $(C_\theta)_{i,j} > 0$ for every $\theta$ outside a zero measure set $Z_{i,j}$. Subsequently, it has to hold that $(\gamma_\theta^*)_{i,j} = 0$ outside the zero measure set $Z \cup Z_{i,j}$ due to (10).

Now, we have $(\gamma_\theta^*)_{i,j} = 0$ or $C_{i,j} = \mathrm{W}^2(\mu_i, \nu_j) = 0$ for some $\theta \in \mathbb{S}^{d-1}$ outside the zero measure set $(Z \cup (\cup_{(i,j)\in\mathrm{Idx}}Z_{i,j}))$. Thus, it holds for almost every $\theta$ that

$$\langle \boldsymbol{\gamma}_\theta^*, C \rangle = 0.$$

Since this expression is an upper bound of $\mathbf{W}^2(\boldsymbol{\mu}, \boldsymbol{\nu}; \mathbb{R}^d)$, it follows that $\mathbf{W}(\boldsymbol{\mu}, \boldsymbol{\nu}; \mathbb{R}^d) = 0$. This concludes the proof since the Wasserstein distance is a positive definite metric. $\square$

**Remark.** Although we stated our statement for the empirical measures employed in our experiments, our proof merely requires that all measures satisfy the Carleman condition. As an example, this would be fulfilled for mixtures of compactly supported measures, i.e., for $\mathcal{P}_e(\mathcal{P}_2(\mathcal{X}))$, $\mathcal{X} \subset \mathbb{R}^d$ compact. Note that $\mathcal{P}_e(\mathcal{P}_2(\mathcal{X}))$ is dense in $\mathcal{P}_e(\mathcal{P}_2(\mathcal{X}))$ (Villani, 2003, Ch. 6). Moreover, Lemma A.1 allows us to show the Lipschitz continuity of DSW on $\mathcal{P}_2(\mathcal{P}_2(\mathcal{X}))$. Combining all of this with statements on continuous extensions of metrics on topological spaces Engelking (1989), we can expect DSW to be a metric on $\mathcal{P}_2(\mathcal{P}_2(\mathcal{X}))$.

### B.2 RELATIONSHIPS BETWEEN METRIC

In this section, we aim to prove our convergence result. We point out that (Han, 2023, Thm. 3.4) contains a proof of weak convergence for measures on general Hilbert spaces. However, the underlying argument appears to rely on an application of a convergence result in infinite-dimensional settings whose validity in this context is, to the best of our understanding, not fully clear.

To prove our convergence statement, we separate the proof into a couple of smaller statements. As a first step, we show that our sliced metrics produce lower bounds for $\mathbf{W}$. We continue with a lemma that relates the subset of $\mathcal{P}_2(L_2([0,1]))$ supported on piecewise constant step functions to Euclidean measures. Based on this, we prove a statement about the equivalence between $\mathbf{DSW}$ and $\mathbf{SW}$. Lastly, we prove a statement about the equivalence between $\mathbf{SW}$ and $\mathbf{W}$, which utilizes the compactness of the support.

First, we state a proposition that allows us to bound $\mathbf{W}$ from below via $\mathbf{DSW}$ and $\mathbf{SW}$.

**Proposition B.3.** *It holds* $C_\xi \, \mathbf{DSW}(\boldsymbol{\mu}, \boldsymbol{\nu}; \xi) \leq \mathbf{SW}(\boldsymbol{\mu}, \boldsymbol{\nu}; \xi) \leq \mathbf{W}(\boldsymbol{\mu}, \boldsymbol{\nu}; \mathbb{R}^d)$ *for* $\boldsymbol{\mu}, \boldsymbol{\nu} \in \mathcal{P}_2(\mathcal{P}_2(\mathbb{R}^d))$ *and* $\xi \in \mathcal{P}_2(L_2([0,1]))$.

*Proof.* DSW is essentially a double integral. For $v \in L_2([0,1])$, $\theta \in \mathbb{S}^{d-1}$ and $\tilde{\gamma} \in \Gamma(\boldsymbol{\mu}, \boldsymbol{\nu})$, the DSW integrand can be estimated using the Cauchy-Schwarz inequality by

$$\mathbf{W}^2(\pi_{v,\sharp}(q_\sharp(\boldsymbol{\pi}_{\theta,\sharp}\boldsymbol{\mu})), \pi_{v,\sharp}(q_\sharp(\boldsymbol{\pi}_{\theta,\sharp}\boldsymbol{\nu})); \mathbb{R}) \leq \int_{\mathcal{P}_2(\mathbb{R}^d) \times \mathcal{P}_2(\mathbb{R}^d)} |\langle q(\pi_{\theta,\sharp}\mu) - q(\pi_{\theta,\sharp}\nu), v \rangle|^2 \, \mathrm{d}\tilde{\gamma}(\mu, \nu)$$

$$\leq \int_{\mathcal{P}_2(\mathbb{R}^d) \times \mathcal{P}_2(\mathbb{R}^d)} \|q(\pi_{\theta,\sharp}\mu) - q(\pi_{\theta,\sharp}\nu)\|^2 \|v\|^2 \, \mathrm{d}\tilde{\gamma}(\mu, \nu).$$

Subsequently, it holds

$$\mathbf{W}^2(\pi_{v,\sharp}(q_\sharp(\boldsymbol{\pi}_{\theta,\sharp}\boldsymbol{\mu})), \pi_{v,\sharp}(q_\sharp(\boldsymbol{\pi}_{\theta,\sharp}\boldsymbol{\nu})); \mathbb{R}) \leq \|v\|^2 \mathbf{W}_2^2(\boldsymbol{\pi}_{\theta,\sharp}\boldsymbol{\mu}, \boldsymbol{\pi}_{\theta,\sharp}\boldsymbol{\nu}; \mathbb{R}) \leq \|v\|^2 \mathbf{W}_2^2(\boldsymbol{\mu}, \boldsymbol{\nu}; \mathbb{R}^d).$$

The last inequality follows from $\mathbf{W}(\pi_{\theta,\sharp}\mu, \pi_{\theta,\sharp}\nu; \mathbb{R}) \leq \mathbf{W}(\mu, \nu; \mathbb{R}^d)$ for $\mu, \nu \in \mathcal{P}_2(\mathbb{R}^d)$ and $\theta \in \mathbb{S}^{d-1}$. Because of $\xi \in \mathcal{P}_2(L_2([0,1]))$, we know that $\int_{L_2([0,1])} \|v\|^2 \, \mathrm{d}\xi(v) = M_2(\xi) < \infty$. Integration with respect to $\xi$ and the uniform measure on $\mathbb{S}^{d-1}$ gives the statement with $C_\xi = 1/\sqrt{M_2(\xi)}$. $\square$

From this relation, we can easily see that $\mathbf{W}(\boldsymbol{\mu}_n, \boldsymbol{\mu}; \mathbb{R}^d) \to 0$ results in $\mathbf{SW}(\boldsymbol{\mu}_n, \boldsymbol{\mu}; \xi) \to 0$ and that $\mathbf{SW}(\boldsymbol{\mu}_n, \boldsymbol{\mu}; \xi) \to 0$ results in $\mathbf{DSW}(\boldsymbol{\mu}_n, \boldsymbol{\mu}; \xi) \to 0$.

In the following statement, we use indicator functions $\mathbf{1}_{x \in S}$ that take the value 1 for $x \in S$ and 0 otherwise.

**Lemma B.4.** *For* $\mu_n, \mu \in \mathcal{P}_2(L_2([0,1]))$ *only supported on fixed-length step functions, i.e., sums of indicator functions of the form*

$$S_P = \left\{ \sum_{i=1}^{\tilde{n}} f_i \mathbf{1}_{x \in P_i} \, : \, f_i \in \mathbb{R} \right\}$$

*for a fixed partition* $\dot{\cup}_{i=1}^{\tilde{n}} P_i = [0,1]$, *and a positive Gaussian measure* $\xi \in \mathcal{P}_2(L_2([0,1]))$, *we have*

$$\mathrm{SW}(\mu_n, \mu; \xi) \to 0 \iff \mathrm{W}(\mu_n, \mu; L_2([0,1])) \to 0.$$

*Proof.* To prove the statement, we want to leverage the established metric equivalence between the Wasserstein and the classical sliced Wasserstein distance on $\mathcal{P}_2(\mathbb{R}^{\tilde{n}})$. Therefore, we construct an isometric mapping $\mathrm{IM}_P : S_P \to \mathbb{R}^{\tilde{n}}$ similar to (Piening & Beinert, 2025a). Instead of considering probability measures on the infinite-dimensional space $L_2([0,1])$, this allows us to consider Euclidean probability measures. We define our $S_P$-isometric mapping on the space $L_2([0,1])$ via

$$\mathrm{IM}_P : L_2([0,1]) \to \mathbb{R}^{\tilde{n}},$$

$$f \mapsto \left( |P_i|^{\frac{1}{2}} \int_{P_i} f(x)\, dx \right)_{i=1}^{\tilde{n}} \in \mathbb{R}^{\tilde{n}}.$$

This is an isometry on $S_p$ because for $f^{(1)}, f^{(2)} \in S_P$ it holds that

$$\|f^{(1)} - f^{(2)}\|^2 = \sum_{i=1}^{\tilde{n}} |P_i|(f_i^{(1)} - g_i)^2 = \|\mathrm{IM}(f^{(1)}) - \mathrm{IM}(f^{(2)})\|^2.$$

Now, this means that

$$\mathrm{W}^2(\mu_n, \mu;\, L_2([0,1])) = \mathrm{W}^2(\mathrm{IM}_{P,\sharp}\,\mu_n, \mathrm{IM}_{P,\sharp}\,\mu;\, \mathbb{R}^{\tilde{n}}). \tag{11}$$

Now, we aim to construct a similar relation for the $\xi$-based sliced Wasserstein distance. We require $\xi_P \in \mathcal{P}_2(\mathbb{R}^{\tilde{n}})$ based on $\xi \in \mathcal{P}_2(L_2([0,1]))$. Therefore, we again employ our mapping to link the $L_2$-projection to a projection on $\mathbb{R}^{\tilde{n}}$. For $f \in S_P$, $g \in L_2([0,1])$, we have $\langle f, g \rangle = \langle \mathrm{IM}_P(f), \mathrm{IM}_P(g) \rangle$. Hence, we define $\xi_P := \mathrm{IM}_{P,\sharp}\,\xi$. As a sliced counterpart of (11), we get that

$$\mathrm{SW}^2(\mu_n, \mu; \xi) = \mathrm{SW}^2(\mathrm{IM}_\sharp\,\mu_n, \mathrm{IM}_\sharp\,\mu; \xi_P).$$

Now, we want to utilize Proposition 3.3 to connect our $\xi$-based sliced Wasserstein to the classical sliced Wasserstein distance. Indeed, since $\xi$ is a Gaussian measure and $\mathrm{IM}_P$ is linear, $\xi_P$ is Gaussian by definition of a Gaussian measure. Moreover, it is a nondegenerate Gaussian since $\xi$ is positive. Thus, $\mathrm{SW}(\mathrm{IM}_\sharp\,\mu_n, \mathrm{IM}_\sharp\,\mu; \xi_P)$ is topologically equivalent to the classical sliced distance $\mathrm{SW}(\mathrm{IM}_\sharp\,\mu_n, \mathrm{IM}_\sharp\,\mu)$ by Proposition 3.3. Since SW and W induce the same weak topology on $\mathcal{P}_2(\mathbb{R}^{\tilde{n}})$ (Nadjahi et al., 2020), we overall conclude that

$$\mathrm{SW}(\mu_n, \mu;\, \xi) \to 0$$
$$\iff \mathrm{SW}(\mathrm{IM}_\sharp\,\mu_n, \mathrm{IM}_\sharp\,\mu; \xi_P) \to 0$$
$$\iff \mathrm{SW}(\mathrm{IM}_\sharp\,\mu_n, \mathrm{IM}_\sharp\,\mu) \to 0$$
$$\iff \mathrm{W}(\mathrm{IM}_\sharp\,\mu_n, \mathrm{IM}_\sharp\,\mu;\, \mathbb{R}^{\tilde{n}}) \to 0$$
$$\iff \mathrm{W}(\mu_n, \mu;\, L_2([0,1])) \to 0.$$

$\square$

**Proposition B.5.** *Given a positive Gaussian $\xi \in \mathcal{P}_2(L_2([0,1]))$ and empirical meta-measures $\boldsymbol{\mu}_n, \boldsymbol{\mu} \in \mathcal{P}_e^N(\mathcal{P}^{\tilde{n}}(\mathcal{X}))$, $\mathcal{X} \subset \mathbb{R}^d$ compact, we have that*

$$\mathbf{DSW}(\boldsymbol{\mu}_n, \boldsymbol{\mu};\, \xi) \to 0 \iff \mathbf{SW}(\boldsymbol{\mu}_n, \boldsymbol{\mu};\, \xi) \to 0.$$

*Proof.* We assume that

$$\mathbf{DSW}^2(\boldsymbol{\mu}, \boldsymbol{\nu};\, \xi) = \int_{\mathbb{S}^{d-1}} \mathrm{SW}^2\left( q_\sharp(\boldsymbol{\pi}_{\theta,\sharp}\boldsymbol{\mu}_n), q_\sharp(\boldsymbol{\pi}_{\theta,\sharp}\boldsymbol{\mu});\, \xi \right) \mathrm{d}\mathbb{S}^{d-1}(\theta) \to 0.$$

It follows that

$$\mathrm{SW}(q_\sharp\boldsymbol{\pi}_{\theta,\sharp}\boldsymbol{\mu}_n, q_\sharp\boldsymbol{\pi}_{\theta,\sharp}\boldsymbol{\mu};\, \xi) \to 0$$

for almost any $\theta \in \mathbb{S}^{d-1}$. Since we are dealing with fixed, uniform weights, all quantile functions are piecewise constant step functions with a fixed step length. By Lemma B.4, it thus follows that

$$\mathrm{W}(q_\sharp\boldsymbol{\pi}_{\theta,\sharp}\mu_n, q_\sharp\boldsymbol{\pi}_{\theta,\sharp}\mu_n;\, L_2([0,1]) = \mathbf{W}(\boldsymbol{\pi}_{\theta,\sharp}\boldsymbol{\mu}_n, \boldsymbol{\pi}_{\theta,\sharp}\boldsymbol{\mu};\, \xi) \to 0$$

for almost every $\theta \in \mathbb{S}^{d-1}$. Since the compact support results in boundedness, it thus follows from the dominated convergence theorem that

$$\mathbf{SW}^2(\boldsymbol{\mu}_n, \boldsymbol{\mu};\, \xi) = \int_{\mathbb{S}^{d-1}} \mathbf{W}^2(\boldsymbol{\pi}_{\theta,\sharp}\boldsymbol{\mu}, \boldsymbol{\pi}_{\theta,\sharp}\boldsymbol{\nu};\, \mathbb{R}) \mathrm{d}\mathbb{S}^{d-1}(\theta) \to 0.$$

$\square$

Now, we state the last result. Note that our proof is inspired by a proof in (Piening & Beinert, 2025b).

**Proposition B.6.** *Given $\boldsymbol{\mu}_n, \boldsymbol{\mu} \in \mathcal{P}_e^N(\mathcal{P}_e^{\tilde{n}}(\mathcal{X}))$, $\mathcal{X} \subset \mathbb{R}^d$ compact, it holds that*

$$\mathbf{SW}(\boldsymbol{\mu}_n, \boldsymbol{\mu}; \xi) \to 0 \iff \mathbf{W}(\boldsymbol{\mu}_n, \boldsymbol{\mu}; \mathbb{R}^d) \to 0.$$

*Proof.* We write

$$\boldsymbol{\mu}_n = \frac{1}{N} \sum_{i=1}^N \delta_{\mu_{n,i}}, \quad \boldsymbol{\mu} = \frac{1}{N} \sum_{j=1}^N \delta_{\mu_j}.$$

Now, we assume that

$$\mathbf{SW}^2(\boldsymbol{\mu}_n, \boldsymbol{\mu}; \xi) = \int_{\mathbb{S}^{d-1}} \sum_{k=1}^N \sum_{\ell=1}^N \gamma_{\theta,k,\ell}^{n,*} \mathbf{W}_2^2(\pi_{\theta,\sharp} \mu_{n,k}, \pi_{\theta,\sharp} \mu_\ell; \mathbb{R}) \, \mathrm{d}\mathbb{S}^{d-1}(\theta) \to 0,$$

where $\boldsymbol{\gamma}_\theta^{n,*} \in \Gamma(\pi_{\theta,\sharp} \boldsymbol{\mu}_n, \pi_{\theta,\sharp} \boldsymbol{\mu}) \subset \mathbb{R}_{\geq 0}^{N \times N}$ denotes the optimal projected WoW plan for a fixed $\theta \in \mathbb{S}^{d-1}$. Because of $L_2(\mathbb{S}^{d-1})$ convergence, for any subsequence of $\boldsymbol{\mu}_n$, we find a further subsequence $(\boldsymbol{\mu}_{n_m})_{m \in \mathbb{N}}$ such that

$$\sum_{k=1}^N \sum_{\ell=1}^N \gamma_{\theta,k,\ell}^{n_m,*} \mathbf{W}_2^2(\pi_{\theta,\sharp} \mu_{n_m,k}, \pi_{\theta,\sharp} \mu_\ell; \mathbb{R}) \to 0 \quad \text{for almost every } \theta \in \mathbb{S}^{d-1} \tag{12}$$

pointwisely. Since $\mathcal{X}$ is compact, $(\boldsymbol{\mu}_{n_m})_{m \in \mathbb{N}}$ can be chosen such that $\mu_{n_m,k}$ converges to some $\tilde{\mu}_k \in \mathcal{P}_2(\mathbb{R}^d)$. Moreover, since $\boldsymbol{\mu}_n$ and $\boldsymbol{\mu}$ are both empirical meta-measures with $N$ support points, we can assume without loss of generality that $\gamma_{\theta,k,\ell}^{n,*}$ is a permutation matrix, i.e., $\gamma_{\theta,k,\ell}^{n,*} \in \{0, \frac{1}{N}\}^{N \times N}$ with only one positive value per row or column (Peyré & Cuturi, 2019, Prop. 2.1).

For any $\theta \in \mathbb{S}^{d-1}$ such that (12) holds true, it follows that $\gamma_{\theta,k,\ell}^{n_m,*} \to \frac{1}{N}$ in the case of $\pi_{\theta,\sharp} \tilde{\mu}_k = \pi_{\theta,\sharp} \mu_\ell$ and $\gamma_{\theta,k,\ell}^{n_m,*} \to 0$ otherwise. It follows that either $\gamma_{\theta,k,\ell}^{\theta,*} \to 0$ or $\mathbf{W}^2(\mu_{n_m,k}, \mu_\ell) \to 0$. Due to the compactness assumption, we know that $\gamma_{\theta,k,\ell}^{n_m,*}$ and $\mathbf{W}^2(\mu_{n_m,k}, \mu_\ell)$ are bounded and thus

$$\mathbf{W}^2(\boldsymbol{\mu}_n, \boldsymbol{\mu}; \mathbb{R}^d) \leq \sum_{k=1}^K \sum_{\ell=1}^K \gamma_{\theta,k,\ell}^{n_m,*} \mathbf{W}^2(\mu_{n_m,k}, \mu_\ell; \mathbb{R}^d) \to 0 \quad \text{as} \quad n \to \infty,$$

Because this holds true for any subsequence of $(\boldsymbol{\mu}_n)_{n \in \mathbb{N}}$, the statement follows. $\qquad\square$

Combining Proposition B.5 and Proposition B.6 gives the second statement of Theorem 4.1.

# C  ADDITIONAL DETAILS AND EXPERIMENTS

## C.1  IMPLEMENTATION DETAILS

All experiments were conducted with *Python* on a system equipped with a 13th Gen Intel Core i5-13600K CPU and an NVIDIA GeForce RTX 3060 GPU with 12 GB of memory. For all experiments concerning the s-OTDD[3], OTDD, and STLB[4], we employ the official implementations for algorithms and experiments based on the corresponding public GitHub repositories. Unless stated in the experimental description, we use the default hyperparameters for algorithms, including the entropic regularization parameter for OTDD computation. For all other Euclidean Wasserstein computations, we employ the *geomloss* package (Feydy et al., 2019) to estimate entropically regularized Wasserstein distances, where we set the entropic '*blur*' parameter to 0.01. All WoW distances are calculated by estimating a pairwise Wasserstein cost matrix using this *geomloss* geomloss and finally solving the unregularized OT problem given this cost matrix using the *POT* package.

For our own implementation, we employ linear interpolation between the two closest support points for our quadrature grid points. Moreover, instead of sampling the Gaussian processes $v$ and the unit directions $\theta$ completely independently, we sample 10 or 100 random Gaussian processes $v$ for each sampled unit direction $\theta$ to reduce the number of quantile computations. For our low-dimensional point cloud experiment and our mid-dimensional patch experiment, we employ 100 random 'outer projections' $\theta$ and 100 random 'inner projections' $v$ per $\theta$ ($10,000$ in total). For the high-dimensional (s-)OTDD experiments, we employ 1000 random 'outer projections' $\theta$ and 10 random 'inner projections' $v$ per $\theta$ (also $10,000$ in total).

## C.2  PRACTICAL GUIDE ON PARAMETER CHOICE

The numerical implementation depends on multiple parameters, namely the kernel parameter $\sigma$, the grid size $R$, and the projection number $S$. We conduct multiple parameter studies that can be found in Sections C.3, C.5, and C.6.

For the parameters $S$ and $R$, we note that both of them are essentially integration parameters of a Monte Carlo integration. The parameter $R$ is the size of the quantile integration grid, and $S$ is the number of Monte Carlo steps. Thus, we would generally expect better results for higher choices of $R$ and $S$. However, the runtime scales linearly with the number of Monte Carlo steps $S$ and Monte Carlo typically converges with a rate of $\mathcal{O}(S^{-1/2})$. Thus, choosing $S$ is mainly a question of balancing performance and runtime, where the marginal value of an additional Monte Carlo projection decreases for high $S$. As for the grid size $R$, we note that the quantile functions we aim to integrate are rather simple. In particular, they are monotonically increasing step functions. In line with this observation, we only found small marginal performance increases for $R > 10$ in most experiments. Moreover, we note that high choices of $R$ result in sampling from highly correlated, high-dimensional Gaussians, which might become numerically unstable for some implementations.

During our experiments regarding SQW for the shape experiment in Section 5.1, we only found a small impact of $\sigma$. This is in line with Proposition 3.2 showing that variations of $\sigma$ result in an equivalent metric. While $\sigma = 0$ works for our discretization in practice and essentially reduces to the slicing approach discussed in (Piening & Beinert, 2025a), it corresponds to sampling a white noise process that is not contained in $L_2([0,1])$. Therefore, $\sigma = 0$ is not covered by our theory. The limit $\sigma \to \infty$ leads to an increasing correlation and to sampling constant test functions. However, as constant functions are not dense in $L_2([0,1])$, the conditions of Theorem 3.1 would not be met anymore. Hence, SQW might lose its metric properties. Thus, we generally recommend choosing $\sigma$ as a value from $[0.001, 0.1]$.

---

[3]Code: `https://github.com/hainn2803/s-OTDD`.
[4]Code: `https://github.com/MoePien/slicing_fused_gromov_wasserstein`.

### C.3 SLICED FUNCTIONAL OPTIMAL TRANSPORT ON $L_2([0,1])$

Both of our introduced sliced distances, SQW and DSW, rely on the $\xi$-based SW distance on $\mathcal{P}_2(L_2([0,1]))$ and its numerical implementation. We study the impact of our parameter choices on this distance by looking at increasingly finer function discretizations $R$ for different kernel parameters $\sigma$. We consider two empirical measure pair in $\mathcal{P}_2(L_2([0,1]))$ defined via

$$f_i^{(1)}(x) = \cos(ix), \ \ h_j^{(1)}(x) = \sin(jx+j\pi), \ \ f_i^{(2)}(x) = \cos(ix+i)+\sin(x), \ \ h_j^{(2)}(x) = \sin(jx)^j$$

and

$$\mu^{(k)} = \frac{1}{5}\sum_{i=1}^{5}\delta_{f_i^{(k)}}, \quad \nu^{(k)} = \frac{1}{10}\sum_{j=1}^{10}\delta_{h_j^{(k)}}, \quad k = 1,2.$$

In Figure 6, we plot our Monte Carlo estimate for the $\xi$-based SW between $\mu^{(1)}$ and $\nu^{(1)}$ resp. $\mu^{(2)}$ and $\nu^{(2)}$ estimates for different Gaussian bandwidth parameters $\sigma$ and equispaced discretization grids with varying size $R$ ($20 \leq R \leq 100$). To investigate the limit case for $\sigma \to 0$, we also include the case of isotropic Gaussian slicing directions $\theta_l \sim \mathcal{N}(\mathbf{0}, \mathbf{I}_N)$ due to $(k_\sigma(y_i, y_j))_{1 \leq i,j \leq N} \to \mathbf{I}_N$. While this limit case is well-defined for finite discretization, the limit process would be a white noise process, i.e., a process with sample paths outside of $L_2([0,1])$. On the one hand, we observe that the SW estimate depends heavily on the grid resolution $R$, especially for isotropic Gaussian directions and for $\sigma$ small. On the other hand, we see that our numerical estimate is less sensitive to the discretization for larger $\sigma$ and remains stable given a sufficiently large grid. To get accurate estimates for our comparison, we employ $S = 10,000$. We plot all employed functions in Figure 5.

In the main paper, we only employed the radial basis function (RBF) kernel $k_\sigma$. However, we point out that we might use other kernels, such as the Brownian motion (BM) kernel $k(s,t) = \min(\{s,t\})$. As an example, we repeat the experiment portrayed in Figure 6 with the BM kernel and plot the results in the last column of Figure 6. Similar to an RBF kernel with high $\sigma$, we see that the resulting sliced distance is invariant to the discretization.

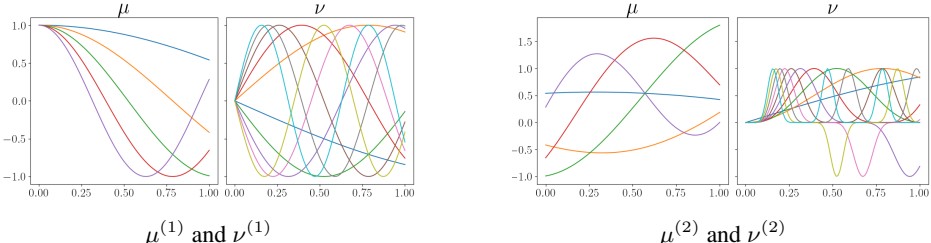

$\mu^{(1)}$ and $\nu^{(1)}$ $\qquad\qquad\qquad\qquad\qquad$ $\mu^{(2)}$ and $\nu^{(2)}$

Figure 5: Plots of the support functions of the two empirical measure pairs from Section C.3.

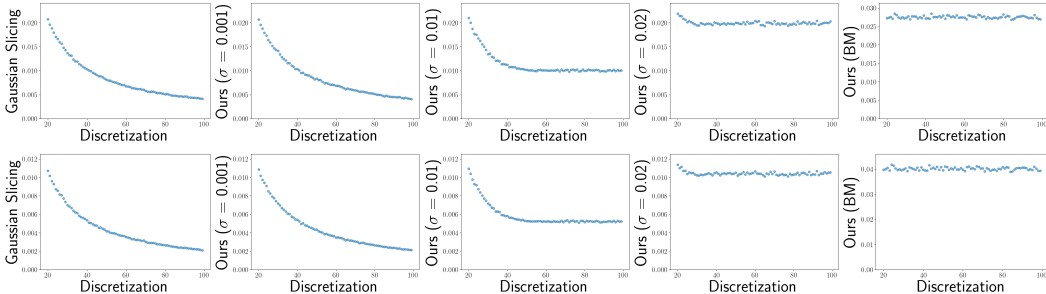

Figure 6: Dependence between function discretization and SW estimates between $\mu^{(1)}$ and $\nu^{(1)}$ (upper row) and $\mu^{(2)}$ and $\nu^{(2)}$ (bottom row) for different kernel choice. Smaller $\sigma$ values lead to less correlated Gaussian slicing directions for the $k_\sigma$ kernel, where we include the limit case of fully uncorrelated Gaussian slicing directions in the first column. For such small $\sigma$, the numerical estimator depends heavily on the discretization.

### C.4 EXTENSION OF SECTION 5.1 ON SHAPE CLASSIFICATION

#### C.4.1 MNIST-2000

MNIST-2000 used in Section 5.1 is a synthetic dataset, which we build based on the training samples of MNIST. For each of the five digits 0–4, we first compute the mean image. From each digit, we generate 20 empirical point clouds consisting each of 2000 points. For this, we sample the mean image proportionally to pixel intensity. Afterwards, we disturb the point clouds by Gaussian noise and apply a random rotation. The points clouds are equipped with Euclidean distance, which are represented by matrices. In total, this yields 20 almost isometric shapes per digit. We show one sample per class in Figure 7.

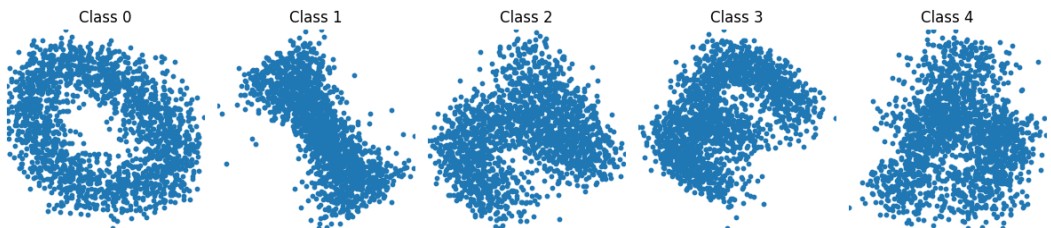

Figure 7: MNIST-2000 Samples.

#### C.4.2 STABILITY ANALYSIS OF CLASSIFICATION PIPELINE

The shape classification experiment presented in Table 1 depends on a random initialization determined by a random seed and on the choice of the $k$-nearest-neighbor ($k$-NN) algorithm. Since each dataset includes only about 100 shapes, cross-validation offers too little statistical support for selecting $k$. For this reason, we fix $k$ a priori. We perform an additional experiment to demonstrate the impact of $k$ and the random seed.

For our main experiment, we choose $k = 3$ instead of $k = 1$ to mitigate the impact of small outliers. Additionally, we aim for a small $k$ due to the limited number of samples per class, e.g., 10 samples per class for the '2D Shapes' dataset. To illustrate the impact of the choice of $k$ and the random initialization, we rerun our classification pipeline for the '2D Shapes' and the 'Animals' dataset with a different initialization and report the resulting accuracies for $k = 1, 3, 5$. The results are presented in Table 2 and are in line with those presented in Table 1 in the main text. We observe a drop in performance for $k = 5$, presumably due to limited data per class.

| k | 1-NN Acc. (%) | 3-NN Acc. (%) | 5-NN Acc. (%) | 1-NN Acc. (%) | 3-NN Acc. (%) | 5-NN Acc. (%) |
|---|---|---|---|---|---|---|
| SQW (Ours) | $99.9 \pm 0.6$ | $99.4 \pm 1.3$ | $98.6 \pm 2.1$ | $99.5 \pm 0.9$ | $99.1 \pm 1.3$ | $68.5 \pm 2.1$ |
| TLB | $100.0 \pm 0.1$ | $100.0 \pm 0.3$ | $99.9 \pm 0.4$ | $100.0 \pm 0.0$ | $100.0 \pm 0.0$ | $70.9 \pm 0.0$ |
| STLB | $99.8 \pm 0.6$ | $99.4 \pm 1.3$ | $98.3 \pm 2.2$ | $99.8 \pm 0.7$ | $99.3 \pm 1.1$ | $69.2 \pm 1.7$ |
| AE | $99.9 \pm 0.4$ | $99.7 \pm 1.0$ | $99.0 \pm 1.9$ | $98.9 \pm 1.4$ | $97.8 \pm 1.9$ | $67.6 \pm 2.3$ |
| GW | $100.0 \pm 0.2$ | $99.8 \pm 0.6$ | $98.9 \pm 1.1$ | $100.0 \pm 0.0$ | $100.0 \pm 0.0$ | $70.9 \pm 0.0$ |
| | (a) 2D Shapes | | | (b) Animals | | |

Table 2: Classification accuracies for the nearest-neighbor algorithm ('$k$-NN Acc.') for varying neighborhood definitions $k$ and a random seed that is different from the one used in Table 1.

## C.5 EXTENSION OF SECTION 5.2 ON DATASET COMPARISON

### C.5.1 RELATION BETWEEN CLASSIFICATION ACCURACY AND DATASET DISTANCES

Originally, the OTDD and the s-OTDD have been introduced to estimate the difficulty of transfer learning. Therefore, we extend the experiment presented in Figure 2 to consider the connection between the different dataset distances and classifier accuracy. As a simple experiment, we consider the accuracy of a 5-nearest-neighbor (5-NN) MNIST classifier as our target. In particular, for each of our 100 MNIST splits, we average the two accuracies of classifying the first split based on the second split and vice versa. Now, this averaged accuracy is our target quantity.

Again, we estimate the correlation between this target accuracy and various dataset distances. Here, we would expect a negative correlation since the dataset splits that are further away from each other should lead to worse training results. Beyond the previously reported OTDD, s-OTDD ($S = 10^5$), and our DSW ($S = 10^5$, 1000 outer projections, 10 inner ones), we additionally report results for the Gaussian OTDD, the s-OTDD with $S = 10^4$), our DSW with $S = 1000$ (100 outer projections, 10 inner ones). Note that the Gaussian OTDD approximates all inner pairwise Wasserstein distances via a Gaussian approximation (Alvarez-Melis & Fusi, 2020). Essentially, this distance reduces to the mixture Wasserstein distance between Gaussian mixtures (Delon & Desolneux, 2020), since we compute all distances without feature costs, i.e, on $\mathcal{P}(\mathcal{P}(\mathbb{R}^d))$ instead of $\mathcal{P}(\mathcal{Y} \times \mathcal{P}(\mathbb{R}^d))$.

Looking at the results in Figure 8, we observe the strongest correlation for the OTDD and the Gaussian OTDD with values of $\approx -0.5$. Regarding the sliced metrics, we observe a stronger correlation for our DSW ($\approx -0.45$) compared to the s-OTDD ($\approx -0.35$). As for the number of projections, both distances seem to be stable with respect to the number of random projections $S$, and we observe only small differences between $S = 10^4$ and $S = 10^5$.

Generally, we observe a clearly negative correlation of approximately $-0.5$ across metrics. While these correlations are overall weaker than in other related experiments (Alvarez-Melis & Fusi, 2020; Nguyen et al., 2025), we attribute this to the small variability of the 5-NN classifier accuracy (ranging from 88% to 94%) and the omission of feature costs. Thus, we hypothesize that the inclusion of the feature cost via hybrid hierarchical slicing (Nguyen & Ho, 2024) would further strengthen these correlations.

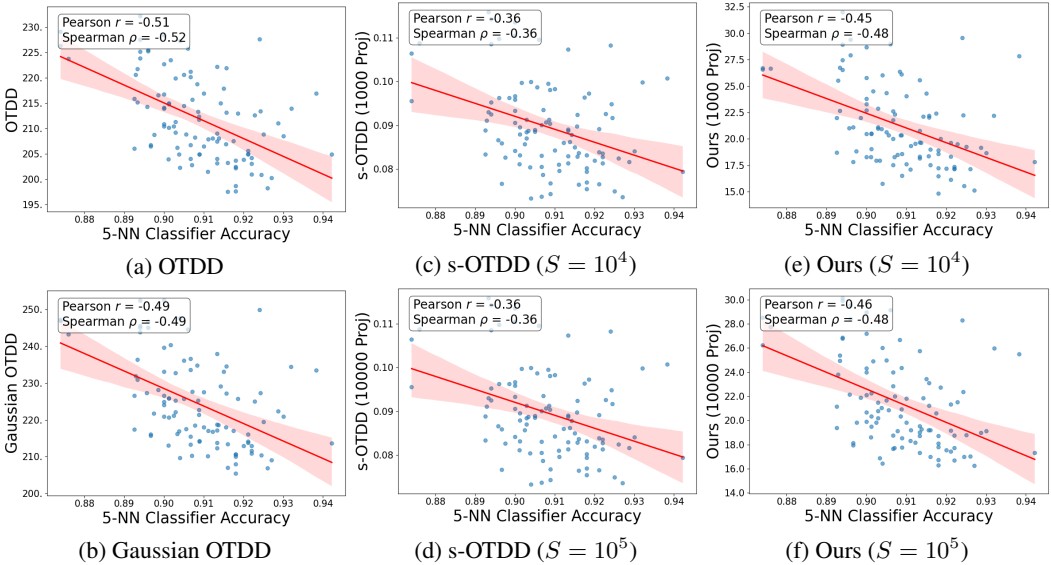

Figure 8: Scatter plots and correlations ($\uparrow$) between the accuracy of a 5-NN MNIST classifier and the OTDD (8a), the Gaussian-OTDD (8b), the s-OTDD with $S = 10^4, 10^5$ (8c, 8d) and DSW ('Ours') with $S = 10^4, 10^5$ (8e, 8f). All distances are computed without a feature cost on the labels.

### C.5.2 STABILITY ANALYSIS OF SLICING PARAMETERS

We investigate the impact of the projection number $S$, grid size $R$ and the kernel parameter $\sigma$ by considering the correlation between DSW and OTDD on MNIST displayed in Figure 2a. As the projection number $S$ is a product of the number of outer and inner projections, see Section C.1, we fix the number of inner projections to $10$ and only vary the number of outer projections. In particular, we simulate the DSW distance for all combinations of $S = 10^3, 10^4, 10^5$, $R = 10, 100, 200$, and $\sigma = 0.01, 0.1$.

For each simulation, we compute the Pearson and Spearman correlation coefficients and present the results in Table 3. All coefficients are approximately in the range from $0.9$ to $0.95$. We see that the number of slices $S$ is by far the most important parameter, whereas the impact of the grid size $R$ and the kernel parameter $\sigma$ is more subtle. In particular, increasing $S$ from $10^3$ to $10^4$ increases the correlation significantly (from $0.9$ to $0.94$), whereas the increase from $10^4$ to $10^5$ only has a slight impact (raising the correlation from $0.94$ to $0.95$).

| $S$ | $R$ | $\sigma$ | Pearson | Spearman |
|---|---|---|---|---|
| 1000 | 10 | 0.1 | 0.9182 | 0.9187 |
| 1000 | 10 | 0.01 | 0.8966 | 0.8908 |
| 1000 | 100 | 0.1 | 0.9365 | 0.9385 |
| 1000 | 100 | 0.01 | 0.9270 | 0.9214 |
| 1000 | 200 | 0.1 | 0.9265 | 0.9283 |
| 1000 | 200 | 0.01 | 0.9063 | 0.9133 |
| 10000 | 10 | 0.1 | 0.9422 | 0.9369 |
| 10000 | 10 | 0.01 | 0.9453 | 0.9424 |
| 10000 | 100 | 0.1 | 0.9400 | 0.9402 |
| 10000 | 100 | 0.01 | 0.9513 | 0.9474 |
| 10000 | 200 | 0.1 | 0.9424 | 0.9392 |
| 10000 | 200 | 0.01 | 0.9536 | 0.9534 |
| 100000 | 10 | 0.1 | 0.9463 | 0.9451 |
| 100000 | 10 | 0.01 | 0.9524 | 0.9510 |
| 100000 | 100 | 0.1 | 0.9444 | 0.9438 |
| 100000 | 100 | 0.01 | 0.9508 | 0.9494 |
| 100000 | 200 | 0.1 | 0.9442 | 0.9427 |
| 100000 | 200 | 0.01 | 0.9533 | 0.9515 |

Table 3: Correlation between our DSW and OTDD (without feature cost) for various configurations on MNIST, extending the results in Figure 2a.

## C.6 EXTENSION OF SECTION 5.3 ON POINT CLOUD COMPARISON

We extend the point cloud experiments from Section 5.3 by adding two experiments as an analysis of the projection number and runtime. In particular, we vary only the number of 'inner' or 'outer' projections per experiment, see Section C.1. In our point cloud experiments, the runtime hinges on the number of shapes ($N$ and $M$) and the discretization of the shapes ($n$ and $m$). For this analysis, we sample only from the 'chair' class without Gaussian noise and $R = 10$. All results are averaged over five runs.

For our first experiment, we set $n = m = 50$ and vary only $N = M = 10, 20, 30, 40, 50, 60, 70, 80, 90, 100$. For each pair of sampled shape sets, we then compute the WoW distance and our DSW distance. To analyze the impact of the projection number, we calculate it with $S = 100$ (10 outer, 10 inner projections), $S = 1000$ (10 outer, 100 inner p.), and $S = 5000$ (10 outer, 500 inner p.). The results are visualized in Figure 9. Note that we observed a rather high variance for WoW runtime in this experiment, generally. As a result, the plotted WoW runtime estimates in Figure 9 vary rather drastically. Nevertheless, we observe a seemingly polynomial runtime increase for WoW in terms of the number of shapes $N = M$, whereas we only we observe a quasi-linear runtime increase for DSW in terms of $N = M$. As for the projection number, we observe a linear runtime increase in terms of $S$. Moreover, we observe a (small) reduction in the variance of the distance estimate for higher $S$.

For our second experiment, we set $N = M = 10$ and vary only $n = m = 100, 200, 300, 400, 500, 600, 700, 800, 900, 1000$. Again, we compute WoW and DSW. For this experiment, we calculate DSW with $S = 1000$ (100 outer, 10 inner projections), $S = 10,000$ (100 outer, 100 inner p.), and $S = 50,000$ (100 outer, 500 inner p.). The results are visualized in Figure 10. We observe similar results as before, i.e., polynomial runtime increase for WoW and quasi-linear runtime increase for DSW.

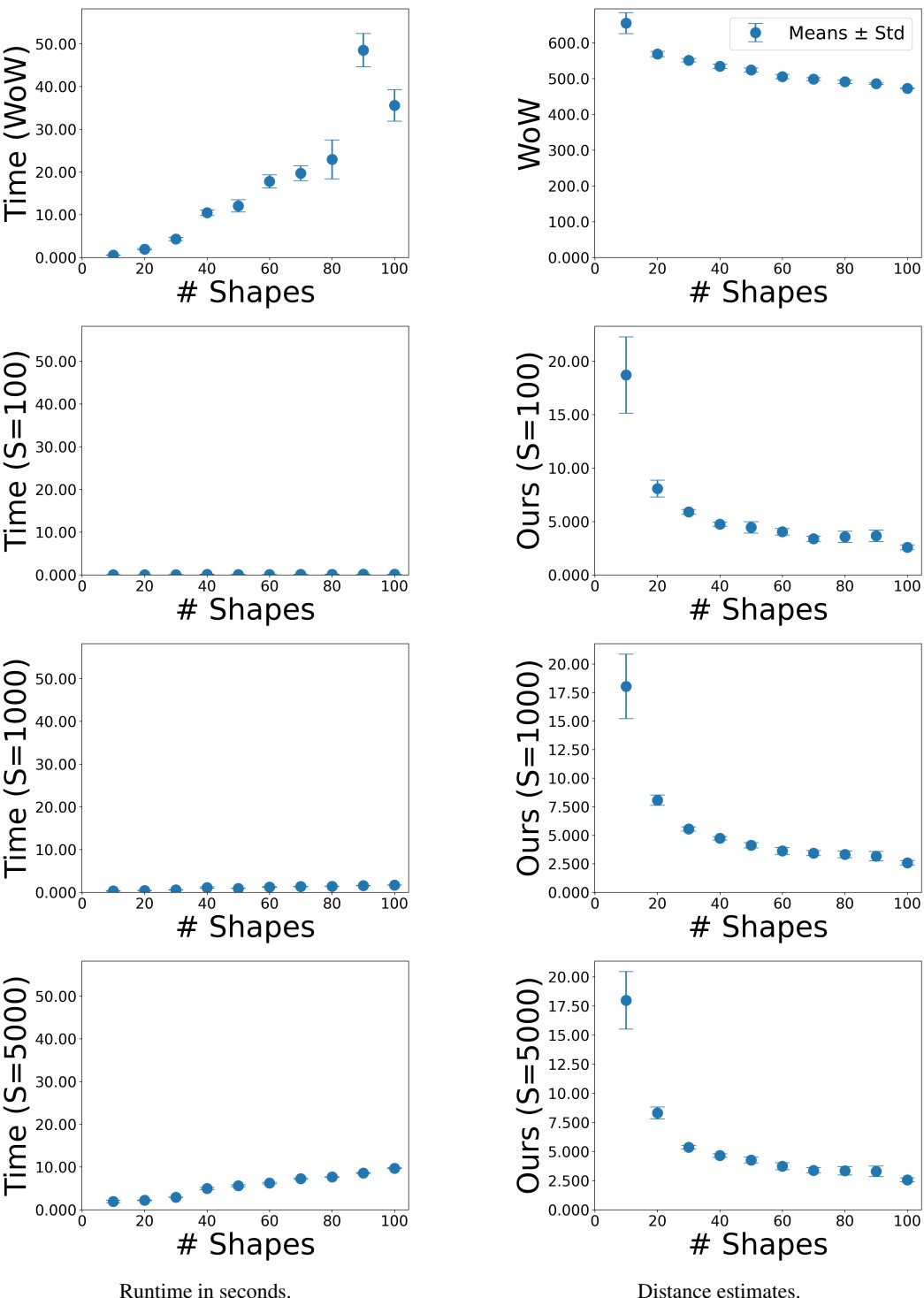

Runtime in seconds.                    Distance estimates.

Figure 9: Averaged WoW and DSW ('Ours') estimates between sets of point clouds for 10 to 100 shapes and projection number $S = 100$, $S = 1,000$, $S = 5,000$.

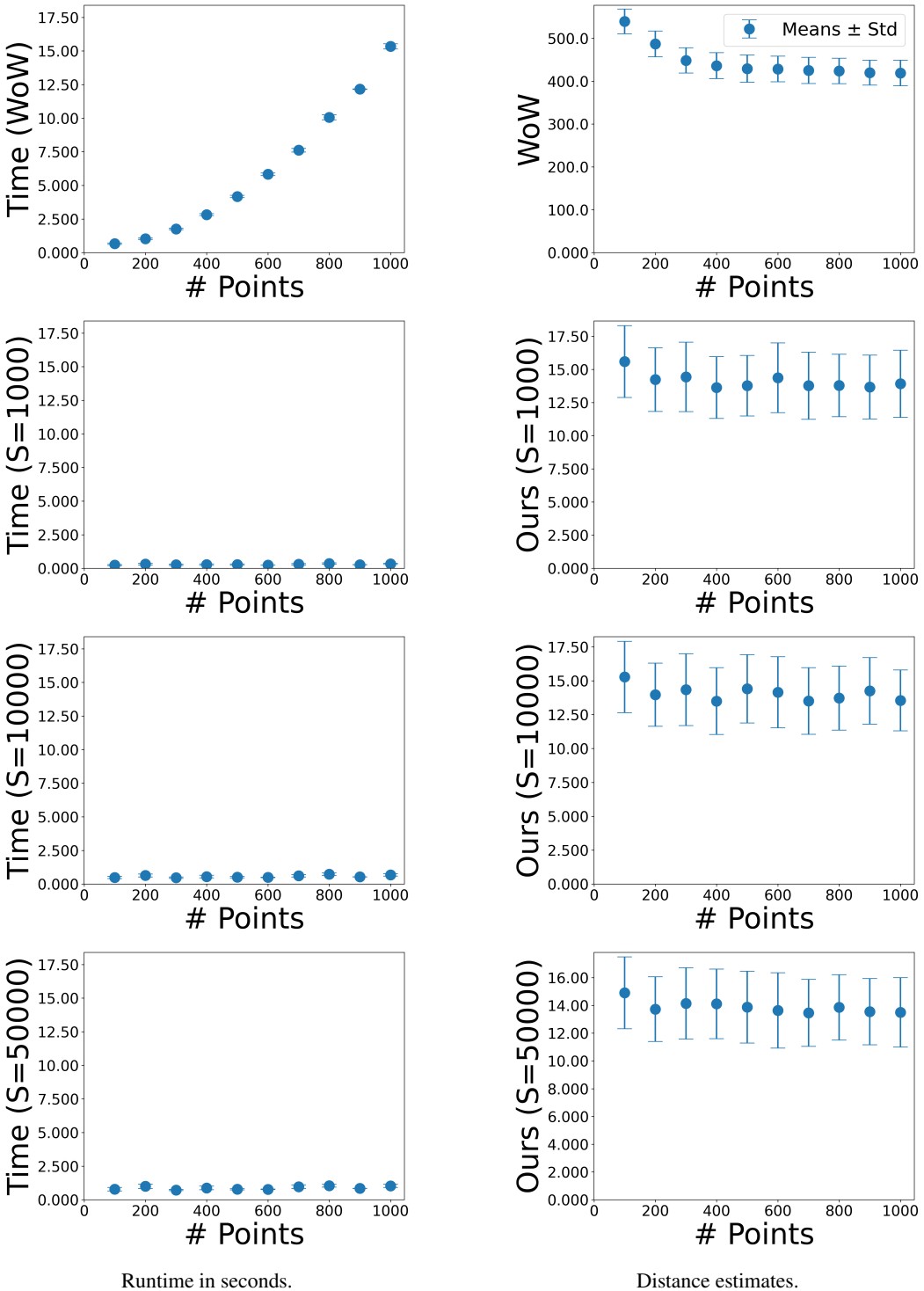

Runtime in seconds.

Distance estimates.

Figure 10: Averaged WoW and DSW ('Ours') estimates between sets of point clouds with 100 to 1000 points per shape and projection number $S = 1000$, $S = 10,000$, $S = 50,000$.

## C.7 Extension of Section 5.4 on Image Comparison via Patches

### C.7.1 Representing Images via Patches

We formalize the patch extraction. For a grayscale image $\text{Img} \in \mathbb{R}^{h \times w}$ define the patch extractor

$$\text{Patch}_k^p : \mathbb{R}^{h \times w} \to \mathbb{R}^{p^2}, \qquad k = 1, \ldots, n_p,$$

with $n_p = (h - p + 1)(w - p + 1)$. Write

$$z_k := \text{Patch}_k^p(\text{Img}) \in \mathbb{R}^{p^2},$$

so that the empirical patch distribution is

$$\mu_{\text{Img}} = \frac{1}{n_p} \sum_{k=1}^{n_p} \delta_{z_k} \in \mathcal{P}_{\text{e}}(\mathbb{R}^{p^2}).$$

Its support is

$$\text{supp}(\mu_{\text{Img}}) = \{\text{Patch}_k^p(\text{Img}) : k = 1, \ldots, n_p\} \subset \mathbb{R}^{p^2},$$

and for a batch $\{\text{Img}_i\}_{i=1}^B$ the meta-measure is

$$\boldsymbol{\mu} = \frac{1}{B} \sum_{i=1}^B \delta_{\mu_{\text{Img}_i}} \in \mathcal{P}_{\text{e}}(\mathcal{P}_{\text{e}}(\mathbb{R}^{p^2})).$$

### C.7.2 The Relevance of Patch Distributions

Many advances in image processing rely on the importance of local image features (Zontak & Irani, 2011). Indeed, convolutional neural networks in computer vision repeatedly apply the same filter to a small receptive field, and this receptive field can be understood as a patch. Moreover, vision transformers decompose images into smaller patches (Dosovitskiy et al., 2021). The advantage of this approach can be motivated by the relevance of small-range dependencies within images.

Notably, a key disadvantage of the standard MSE in imaging is its vulnerability to 'small' image operations. A small shift of all pixels can lead to the explosion of the MSE. However, such operations have only a small effect on the patch distributions. Indeed, the same goes for the translation of an object within an image, see visualizations in (He et al., 2024). In addition, the perceived style of an image seems to be inherently linked to certain localized image features. As an example, style transfer algorithms successfully capture certain artistic aspects of painting via such features (Gatys et al., 2016). Moreover, texture images are characterized by a certain type of stationarity, where a model can generate texture images by simply matching the patch distribution of a single exemplary texture image (Houdard et al., 2023). This could explain why patch-based WoW-type methods lead to clearer discriminiation than Euclidean Wasserstein methods, cf. Figure 12.

### C.7.3 Additional Experimental Details

In the experiment from Section 5.4, we compare distributions over synthetic texture images. We visualize samples from our random Perlin texture model (Perlin, 1985) in Figure 11. Note that our images with varying lacunarity (11a) are all generated with the following Perlin parameters: persistence of 1, scale of 100, 6 octaves. The generation model will be released with the code. For our images with varying persistence (11b), we use different Perlin parameters: lacunarity of 2.5, scale of 100, 5 octaves. Note that while the resulting images in Figure 11a and Figure 11b look rather similar, the ones from Figure 11a display a higher blur and less high-frequency artifacts.

Moreover, we extend Figure 4. In addition to the Wasserstein distance between images represented as Euclidean points and our patch-based DSW distance plotted in the original Figure 4, we present the extended Figure 12 by adding the patch-based WoW distance and the '*Kernel Inception Distance*' (KID) between the distributions of texture images. The patch-based WoW distance is computed on the same patch meta-measures as our patch-based DSW distance. The KID is based on the latent space of a pretrained neural network, see (Sutherland et al., 2018). We see that the DSW and the WoW distance lead to similar results. Also, both are aligned with the KID.

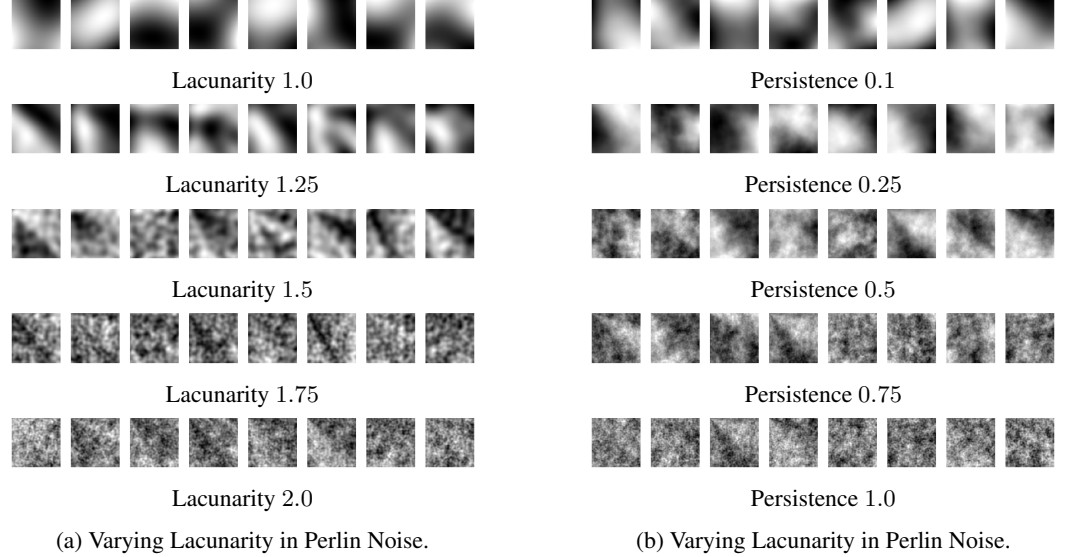

(a) Varying Lacunarity in Perlin Noise.    (b) Varying Lacunarity in Perlin Noise.

Figure 11: Samples from our Perlin texture noise for varying lacunarity (11a) and 'persistence' (11b).

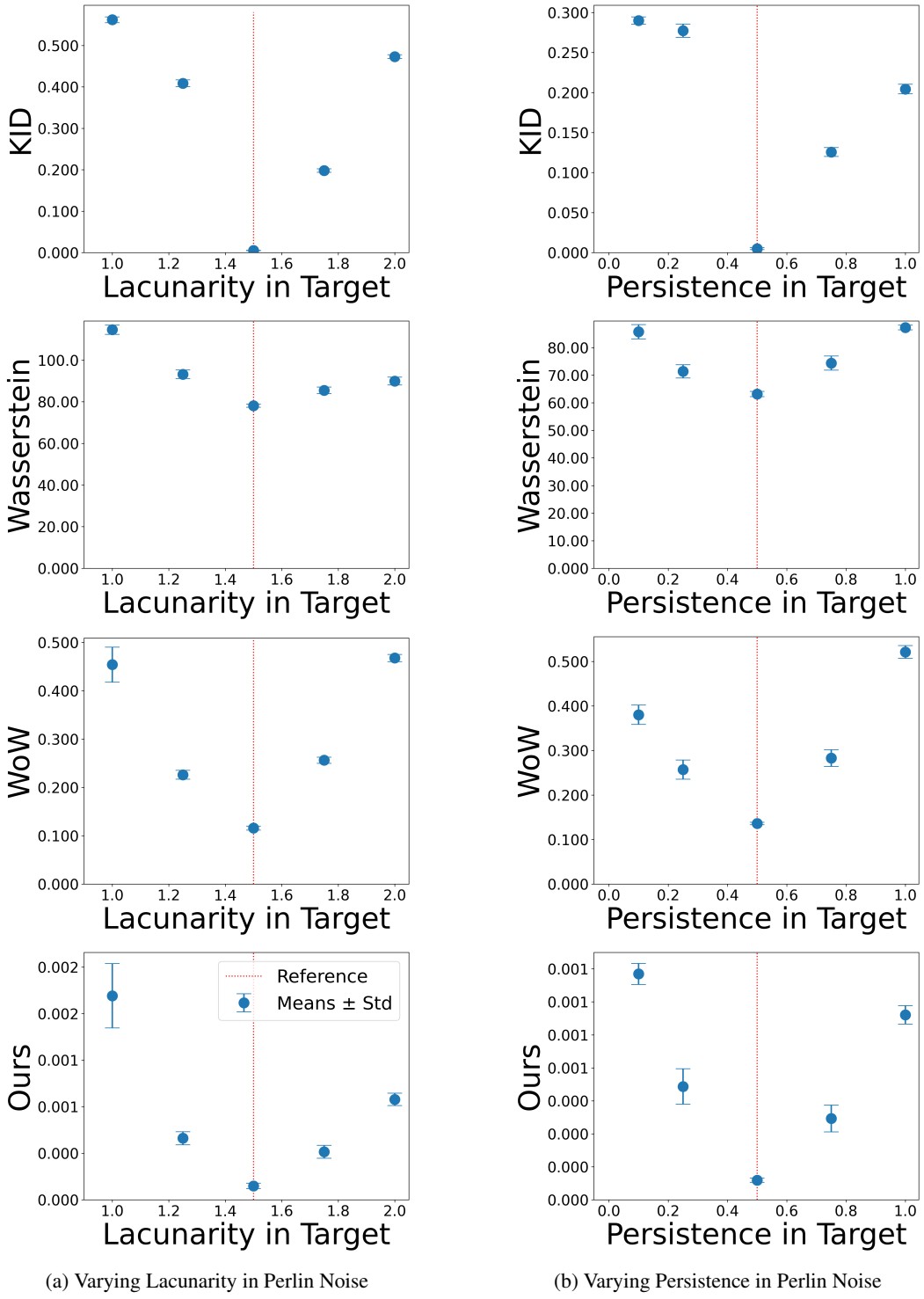

(a) Varying Lacunarity in Perlin Noise

(b) Varying Persistence in Perlin Noise

Figure 12: Comparing synthetic texture image batches via Euclidean Wasserstein, patch-based DSW ('Ours'), patch-based WoW, and KID for varying 'lacunarity' (4a) and 'persistence' (12b).

# D  USE OF LLMS

LLMs have been used to a limited extent to improve grammar and wording in this paper.

