# OpenReview forum: "Slicing Wasserstein over Wasserstein via Functional Optimal Transport"
_ICLR.cc/2026/Conference — ICLR 2026 Poster_

### Official Review · Reviewer_1W1P · 2025-10-24

**Soundness:** 3
**Presentation:** 3
**Contribution:** 3
**Rating:** 8
**Confidence:** 5

**Summary:**

The paper introduces a new computationally efficient way to compare distributions of distributions using sliced optimal transport (SOT). Traditional Wasserstein over Wasserstein (WoW) distances are powerful but computationally expensive and unstable when applied to high-dimensional or hierarchical data. To overcome this, the authors propose the Double-Sliced Wasserstein (DSW) metric, which generalizes sliced Wasserstein distances to arbitrary Banach spaces. Leveraging the isometry between one-dimensional Wasserstein and
$L_2([0,1])$ spaces, they define infinite-dimensional slicing using Gaussian-process–based projections, followed by classical Euclidean slicing. Theoretically, DSW is shown to be topologically equivalent to WoW for discretized meta-measures, offering similar accuracy with far less computational cost. Experiments on datasets, 3D shapes, point clouds, and image distributions demonstrate that DSW achieves comparable or superior results to existing WoW and sliced-OT methods, making it a scalable and robust replacement for hierarchical OT computations.

**Strengths:**

1. The paper rigorously builds upon optimal transport theory, generalizing sliced Wasserstein distances to arbitrary Banach spaces and proving metric properties and equivalence to the original WoW distance.

2. The proposed Double-Sliced Wasserstein (DSW) metric is conceptually original, combining functional (infinite-dimensional) slicing via Gaussian processes with standard geometric slicing, which is both elegant and theoretically justified.

3. DSW substantially reduces the cost of computing hierarchical OT distances compared to WoW and OTDD, making it scalable to large and high-dimensional datasets.

4. By avoiding moment-based approximations (used in previous sliced WoW methods), DSW eliminates numerical instability issues that occur when high-order moments are poorly estimated.

5. Demonstrations across domains (shape analysis, dataset comparison, point clouds, and image distributions) show that the method is general-purpose and effective for both geometric and perceptual tasks.

6.  Experiments are well-chosen and show DSW’s performance to be comparable to or better than established baselines, often with much lower runtime.

**Weaknesses:**

1. While equivalence to WoW is shown for discretized measures, the paper lacks deeper analysis of approximation error, sample complexity, or convergence rates of Monte Carlo integration in DSW. However, these investigation seems to be challenging which might be quite beyond of the scope of the paper.

2.  DSW requires choices such as the Gaussian process kernel width, number of projections (S, R), and quadrature parameters, yet sensitivity analysis or principled tuning guidelines are limited.

3. Despite lower asymptotic cost, the implementation still involves multiple nested integrations (spherical and functional), which may be nontrivial to optimize or parallelize in practice.

**Questions:**

1.  The approach is closely related to  sliced Wasserstein embedding (see Section 4.6 in [1]) which exploits the relationship between sliced Wasserstein space and $L_2$ function space. There are some related aspect should be discussed. For example, instead of double-slicing, I can define SW over SW embeddings (a function or vectors if quantization is used).

2. Why $L_2([0,1])$ is used? As in SW embedding, we can map to $L_2(\mathbb{R})$ using a fixed reference function.

3. A minor comment is that the acronym “DSW” overlaps with “Distributional Sliced Wasserstein” in [2], which also involves modifying the slicing distribution.

[1] An Introduction to Sliced Optimal Transport, Khai Nguyen.

[2] Distributional Sliced-Wasserstein and Applications to Generative Modeling, Nguyen et al

---

> ### Author Response · Authors · 2025-11-20
> **Rebuttal**
>
> Many thanks for reviewing our manuscript, for the positive scores, and for your constructive feedback! We revised our manuscript according to your suggestions and tried to incorporate your comments. Please find our detailed answers below.
>
> **Weakness 1:**
> In the submitted version,
> we have only analyzed the Monte Carlo integration
> for SW on Banach spaces (4)
> in Proposition A.6,
> where we encounter the usual convergence rate of
> $\mathcal O(1 / \sqrt S)$.
> For the fully discrete DSW,
> the analysis is much more involved
> since we also have to handle
> the approximation error
> caused by the quadrature scheme.
> This analysis, however, requires a more in-depth study,
> and we probably have to restrict the considered meta-measures
> to ensure that their quantile functions
> are contained in a suitable function space.
> Ignoring the approximation error,
> we would get a rate of $\mathcal O(1 / \sqrt S)$,
> and we conjecture that this rate can also be achieved
> for our discrete DSW.
>
> **Weakness 2:**
> We added Section C.2,
> a 'practical guide on parameter choice' and extended our previous sensitivity analysis.
> The impact of the parameters is now investigated in Figures 1, 6, 8, 9, 10, and Table 3.
>
> **Weakness 3:**
> We agree that
> our DSW distance relies on multiple integrations.
> For the numerical implementation,
> please notice that
> many steps of DSW may be performed in parallel.
> For instance,
> the calculations for each slicing direction
> could be executed independent of each other.
> Furthermore,
> within each direction,
> the calculation of the sliced meta-measure
> and the computation of its quantile functions
> can be parallelized.
>
> **Question 1:**
> In the revised manuscript,
> we mention the mapping into a function space by the sliced Wasserstein embedding [1] in the introduction
> and briefly discuss the relation on page 6.
> From our side of view,
> the push-forward $q_\sharp \boldsymbol \pi_{\theta,\sharp}$
> can be interpreted as a generalization of the sliced-Wasserstein embedding
> to meta-measures.
>
> **Question 2:**
> For us,
> the main motivation
> to use $L_2([0,1])$
> has been to keep the notation simple
> and to avoid a further reference measure.
> $L_2([0,1])$ is the special case of $L_2(\mathbb R; \rho)$
> where $\rho$ is the uniform distribution on $[0,1]$.
> Theoretically,
> we may use other references $\rho$,
> which then leads to equivalent representations.
> An advantage of $L_2([0,1])$ is, however, the compact support,
> which is numerically easier to handle
> than functions with full support on $\mathbb R$.
>
> **Question 3:**
> Thanks for pointing this out.
> We added a footnote to clarify this overlap with [2].

---

> > ### Comment · Reviewer_1W1P · 2025-11-21
> >
> > Thank you for the reply. All my concerns are addressed. I will keep my positive score. Overall, this is a nice paper which extends sliced Wasserstein to distributions over functions (or distributions). The paper can become stronger if the authors can add a new application which favors this specific construction of distance. However, the paper is sufficient in the current form.

---

> > > ### Author Response · Authors · 2025-11-21
> > > **Thanks**
> > >
> > > Thank you for the swift reply! We are very glad about your positive endorsement.

---

### Official Review · Reviewer_qqXT · 2025-10-28

**Soundness:** 4
**Presentation:** 3
**Contribution:** 3
**Rating:** 6
**Confidence:** 3

**Summary:**

In this paper the authors propose an alternative to the Wasserstein
over Wasserstein (WoW) distance between measures over measures. This amounts of
slicing once in the original space all the distributions and then slicing a
second time the resulting 1D distribution in the set of quantile functions using
directions corresponding to 1D Gaussian distributions. The resulting approximation
is called the Double Sliced Wasserstein (DSW) distance. The authors also show
that DSW can be computed efficiently and converges to 0 similarly to WoW.
Numerical experiments on shape classification, dataset distances, distributions
of point clouds and image datasets seen as distributions of patches show the
potential of the proposed approach.

**Strengths:**

+ The paper is well written and clear.

+ It provides an efficient way to compute similarity between meta distributions,
  and will scale much better to large distributions than existing approaches.

+ Nice theoretical results and discussion on slicing Banach space and 1D
  Wasserstein space.

+ Interesting experiments on meta distribution learning tasks that suggests that the
  proposed approach can be interesting in practice.

+ The method can be directly used to approximate the Third Lower Bound (TLB) of
  Memoli which is a very interesting application on challenging data.

**Weaknesses:**

+ The paper is missing a sensitivity analysis to important parameters such as S
  (number of slices in the original space) and R (number of slices in the
  quantile space) or even the variance $\sigma$ in the Gaussian kernel. This is
  important to understand the tradeoff between computational time and accuracy
  of the approximation. Illustrating variances over multiple runs would also be
  useful since the method is stochastic and Monte Carlo based approximation is
  done here twice and in very large spaces.

+ Application to shape distributions in 5.1 is very nice but could be made
  stronger. Why is K=3 chosen for KNN? For a fair comparison, K should be
  cross-validated for each metric. Also how correlated in DSW to TLB? to GW?
  This is an efficient alternative so how close it is to other metrics is
  important. Also the computational time is actually a bit underwhelming where
  DSW is faster than TLB only on 2 of the 4 datasets. The authors should discuss
  this.

+ Experiments for OTDD in 5.2 are rather limited and do not focus on what is
  very importante for OTDD, which is the estimation of the performance gap
  between models trained on different datasets. It is possible that the proposed
  approach actually works better than OTDD or s-OTDD in this context but this is
  not investigated. Also R=10 is very small for slicing an infinite dimensional
  space, and the method could be more correlated with larger R but this was not
  investigated.

**Questions:**

Please address the weakness comments above and the following questions:

+ In Table 1, what does "Time" refer to? Is it the average time to compute the
  distance between a pair of meta distributions? In this case could you also report
  the variance?

+ Could you be more consistent and use DSW everywhere instead of "Ours"?

+ The proposed Gaussian slicing is important but could faster approach work well
  (theoretically or in practice)? Could we use 0 variance, which is the step
  function as a more efficient a basis not requiring quadrature ?

+ On OTDD experiments It would be interesting to see how DSW compares to other
  scalable OT methods such as OT on GMM where the components of the mixture can
  be estimated on each of the sub-distributions (e.g. Delon 2020 "A
  Wasserstein-type distance in the space of Gaussian Mixture Models").

---

> ### Author Response · Authors · 2025-11-20
> **Rebuttal**
>
> Many thanks for reviewing our manuscript, for the positive scores, and for your constructive feedback! We revised our manuscript according to your suggestions and tried to incorporate your comments. Please find our detailed answers below.
>
> **Weakness 1:**
> In the revised paper,
> we extended our existing experiments with further sensitivity studies. The parameters $S$, $R$ and the Gaussian process parameter $\sigma$ are jointly investigated in the table contained our new Figure 1 and in the supplementary Table 3. The interplay between $\sigma$ and $R$ is studied in Figure 6. The parameter $S$ is further studied in the supplementary Figures 8, 9 and 10.
> Variances over multiple runs are visualized in Figures 3 and 4. Moreover, we repeat the experiment in Table 1 with a new random seed and display the results in our new supplementary Table 2.
>
> **Weakness 2:**
> Please notice that, in section 5.1,
> we focus on small datasets with around 100 samples, since we want to compare the pairwise distances also for time-consuming distances like GW.
> This results in small class sizes
> such that larger cross-validation studies are unavailable.
> To compensate this, we repeated the experiment with $k=1, 3, 5$
> based on a different random initialization.
> The new results are presented in Table 2.
> Moreover,
> we added Appendix C.4.2,
> where we explained our choice for $k$.
>
> To highlight the computational benefits,
> we added a larger experiment to Table 1 (`MNIST-2000'),
> where we see a 6x speed-up over TLB.
> Additionally, we computed the correlation between our SQW and our TLB distance in our new Figure 1.
> We see a very strong correlation of 0.99 - 1.0, demonstrating the value of our approach.
>
> **Weakness 3:**
> To overcome these limitations, we added Section C.5.1, where we investigate the correlation between our DSW and the classification accuracy of a simple 5-nearest-neighbor classifier on MNIST. Indeed, we see that our DSW reaches similar levels of correlation as the OTDD and far stronger than the s-OTDD, see Figure 8.
>
> Please note that $R$ refers to the grid of the employed quadrature scheme.
> In our experiments,
> the quantile functions are rather smooth and monotonously increasing functions.
> The integration of these can be computed sufficiently well
> by setting $R$ to 10 or 100.
> To study the impact of $R$,
> we add several parameter studies highlight the approximation quality of our Monte Carlo estimate, see Figures 1, 6 and Table 3.
>
> **Question 1:**
> This is correct, we mean the averaged time to compute a single distance. We clarified this in the main text. Moreover, we reran the experiments to record the standard deviation in Table 1.
>
> **Question 2:**
> In the revision, we made this more consistent
> by adapting the main text and the captions.
>
> **Question 3:**
> From the theoretical point of view,
> our results are not restricted to the Gaussian slicing
> and also hold for other reference measures.
> The main reason for choosing Gaussian references has been that
> we can easily generate samples.
> From the practical side,
> other processes may also work as well.
> In our current setting,
> however,
> even the step function would not completely overcome the quadrature scheme
> since we still have to compute inner products.
> Moreover,
> the step function would restrict us to a single slicing direction
> violating the conditions of Thm. 3.1.
> The idea to use a more efficient reference process
> is interesting
> but requires further studies.
>
> **Question 4:**
> Computing the Gaussian approximation of the OTDD (`Gaussian OTDD') without feature cost results in the 'Mixture Wasserstein' (MW) distance described by Delon and Desolneux.
> However, the
> MW distance can be slow in high dimensions, as pointed out in the s-OTDD paper (Nguyen et al., 2025).
> For comparison,
> we added a new experiment
> that includes Gaussian OTDD/MW,
> see Figure 8.

---

> > ### Comment · Reviewer_qqXT · 2025-11-21
> > **Thank you**
> >
> > I want to thank the authors that addressed most of my concerns and made the paper stronger. I will change my score to 8.

---

> > > ### Author Response · Authors · 2025-11-21
> > > **Thanks**
> > >
> > > Thanks for getting back to us so quickly! We are delighted about your positive feedback.

---

### Official Review · Reviewer_ir3L · 2025-10-29

**Soundness:** 4
**Presentation:** 4
**Contribution:** 4
**Rating:** 8
**Confidence:** 4

**Summary:**

The objective of this work is to study Optimal Transport (OT) on measures over measures, so called meta-measures, aiming for computational efficiency. As noted by the authors, "a key feature of OT is its applicability to non-Euclidean spaces, even allowing the definition of Wasserstein distances on Wasserstein spaces". This leads to the Wasserstein over Wasserstein (WoW) distance, which is, however, computationally very costly.
Another important existing dissimilarity measure in this "multilevel OT" paradigm is the OT dataset distance (OTDD). OTDD was developed to quantify similarities between labeled datasets in a model-agnostic manner, making it particularly useful for transfer learning. However, OTDD is also highly expensive to compute, and slicing-based variants (s-OTDD) have been proposed to mitigate this. In addition, in applications such as evaluating point cloud generative models, the OT nearest-neighbor accuracy (OT-NNA) test is widely used, but it is computationally demanding too. Also, the Gromov-Wasserstein distance falls under this underlying framework, by treating metric measure spaces as measures over measures. However, one more time, its computation is contly and also it is a non-convex optimization.

In response, the authors propose a general sliced Wasserstein framework for measures on Banach spaces. The theoretical foundation relies on the isometry between the one-dimensional Wasserstein distance and L2([0,1]) via the quantile mapping, as well as slicing techniques in Hilbert spaces (generalizing the inner product to Banach space by utilizing the pairing with linear functionals on its dual space). This allows them to introduce first the slice-quantile WoW (SQW) metric, which roughly speaking corresponds to 1d meta-measures, and then, more importantly, the double-sliced Wasserstein distance (DSW). On the practical side, for particularly sampling, they employ parameterized Gaussian-processes. They demonstrate that DSW can serve as an efficient surrogate for the previously mentioned meta-measures dissimilarities, such as,  WoW, OTDD, s-OTDD, OT-NNA, and GW.

**Strengths:**

- The paper is very well organized.
- Previous works are clearly discussed.
- The aims and problems are stated in a clear way both from the theoretical and practical perspectives.
- The results, formulated as theorems, propositions, and corollaries, highlight important properties of the proposed dissimilarities. For the generalization of the sliced Wasserstein (SW) distance in Banach spaces, since the unit ball is no longer compact in the infinite-dimensional setting, arbitrary probability measures are considered over the dual space of linear functionals and comparisons are then provided: Proposition 3.2 shows that the definition of the new SW is independent of the choice of measure, while Proposition 3.3 establishes that it is a genuine generalization of the classical SW. Other results demonstrate that the proposed dissimilarities are in fact rigorous metrics. Finally, comparisons between the proposed DSW and WoW are presented, showing weak equivalence.
- Comparisons/Experiments with existing meta-measure dissimilarities are presented across different modalities (datasets, shapes, images).

**Weaknesses:**

- A dynamical perspective on the proposed methods is not discussed.
- Sketches of the proofs of the theorems, propositions, and corollaries are not included in the main text, which would provide the reader with at least a vague idea of the technicalities and main steps needed.
- Section 5.1 could benefit from additional comparisons, for example by adding a few more columns in Table 1.
- A discussion of possible reasons explaining the clear discrimination achieved by the proposed distance, as opposed to more traditional methods (Section 5.4, Figure 3), would be interesting.

**Questions:**

- No clousure in span(v) in Thm 3.1 is needed?
- I would appreciate if the authors write the precise definition of $\mathcal P_2(U^*)$.
- Proposition 3.3: I am confused between $\eta$ and the uniform measure $d\mathbb S^{d-1}(\theta)$.

**Details Of Ethics Concerns:**

No concerns.

---

> ### Author Response · Authors · 2025-11-20
> **Rebuttal**
>
> Many thanks for reviewing our manuscript, for the positive scores, and for your constructive feedback! We revised our manuscript according to your suggestions and tried to incorporate your comments. Please find our detailed answers below.
>
> **Weakness 1:**
> Indeed, the WoW distance allow for a dynamic formulation (Haviv et al., 2025; Bonet et al., 2025). However, unlike classical Wasserstein distances, sliced Wasserstein distances are not geodesic anymore, see (Park & Slepčev, 2025). Therefore, we would not expect a straightforward extension of our distances to a dynamical setting. Thus, some extra work is needed for a dynamical extension. Recent approaches enable this by integrating 1d transport plans, see (Liu et al., 2025; Tanguy et al., 2025),
> or selecting a particular 1d transport plan, see (Mahey et al., 2024). In particular, the last approach has been adopted for generative modeling via flow matching (Chapel et al., 2025). We added such extensions as a future research direction to our conclusion.
>
> **Weakness 2:**
> Thanks for this input.
> To improve accessibility, we added brief comments about important cornerstones to the Propositions/Theorems.
> Unfortunately, due to space limitations,
> we could not add more detailed sketches.
>
> **Weakness 3:**
> To highlight the advantage of our approach for large-scale shapes and point clouds, we added an experiment on point clouds with 2000 points to Table 1. Beyond classification, we further investigated the impact of the different numerical parameters and the relation with the TLB. We see a very strong correlation between our SQW and the TLB (0.99-1.0), further strengthening our belief that our SQW is a highly suitable replacement for the TLB, see Figure 1.
>
> **Weakness 4:**
> A key finding in image processing has been that images can be very well-described by small local image features (Zontak & Irani, 2011). Such features can be extracted in various ways, but patch extractors are one of the most popular approaches (Piening et al., 2024).
> Indeed, one can generate relatively realistic images via the minimization of a Wasserstein distance between the patch distributions of a real and synthetic image (Elnekave & Weiss, 2022). Such approaches are especially suitable for texture images (Houdard et al., 2023). As our experiment is dealing with texture images, this could explain the clear discrimination. We added this discussion to our new Appendix C.7.2.
>
> **Question 1:**
> In Thm 3.1,
> we do not need the closure of $\mathrm{span} ~ v$
> because we only consider the span of finitely many elements.
> In our case,
> the span is the line through $v$ and the origin,
> which is closed by itself.
>
> **Question 2:**
> We added the definition of
> $\mathcal P_2(U^\ast) = ( \zeta \in \mathcal P \mid \int_{U^\ast} \lVert v \rVert^2 \mathrm d \zeta(v) < \infty )$
> as well as for $\mathcal P_2(U)$ and $\mathcal P_p(\mathcal X)$.
>
> **Question 3:**
> To clarify Prop. 3.3,
> we added the reference-based SW distance $\mathrm{SW}(\mu,\nu;\eta)$
> regarding the standard Gaussian $\eta$.
> Moreover,
> we added a short note
> about the equality of $\mathrm{SW}(\mu,\nu;\eta)$
> and the classical SW distance on $\mathbb R^d$
> before the proof.

---

> > ### Comment · Reviewer_ir3L · 2025-11-20
> >
> > I thank the authors for their answers, clarifications and further development of the article.

---

> > > ### Author Response · Authors · 2025-11-21
> > > **Thanks**
> > >
> > > Thanks for the quick reply, and we are really glad to read your positive feedback!

---

### Official Review · Reviewer_HPNJ · 2025-10-30

**Soundness:** 3
**Presentation:** 3
**Contribution:** 3
**Rating:** 6
**Confidence:** 3

**Summary:**

The paper introduces a general sliced Wasserstein framework for arbitrary Banach spaces using quantile functions in the function space $L_2([0,1])$. Building on this, the authors present the sliced quantile WoW and double-sliced WoW distance.  Theoretical properties are presented to support the proposed framework.

**Strengths:**

- The paper is well-written and easy to follow. The writing flow is clear and presents the relevant work in a nice and elegant way
- The theoretical results are solid, and the proof in the appendix is organized
- Extending sliced OT to general Banach spaces via quantile maps is interesting and potentially impactful

**Weaknesses:**

- The sliced Wasserstein distance defined in Eq.(4) - there are no experimental results on this distance. How does it compare to other sliced Wasserstein distance empirically?
- SQW’s accuracy gains over competing methods appear modest, aside from FAUST-1000, performance is largely on par with STLB despite comparable runtime. It seems that the proposed method may be beneficial for large-scale datasets, but it is hard to judge based on merely one large dataset that is evaluated.

**Questions:**

- How sensitive are results to the number of projections of DSW in the experiment of optimal transport dataset distance? For example, using 500 projects in s-OTDD varies from using 10000 projections. What would be the correlation in DSW using 500, 1000, or 5000 projections?
- While it can be considered as concurrent work, it would be beneficial if the authors could compare their method conceptually with the work Busemann Functions in the Wasserstein Space: Existence, Closed-Forms, and Applications to Slicing. What are the key differences?
- How does the proposed method perform in transfer learning tasks?

---

> ### Author Response · Authors · 2025-11-20
> **Rebuttal**
>
> Many thanks for reviewing our manuscript,
> for the positive scores,
> and for your constructive feedback!
> We revised our manuscript according to your suggestion
> and tried to incorporate your comments.
> Please find our detailed answers below.
>
> **Weakness 1:**
> Our main motivation to introduce
> the $\xi$-based SW distance in (4)
> has been to generalize the classical SW distance
> to infinite-dimensional linear spaces,
> especially to $L_2([0,1])$.
> This slicing is then used as
> key ingredient for $\mathbf{SQW}$ and $\mathbf{DSW}$.
> Regarding applications on general measures over $L_2([0,1])$,
> the $\xi$-based SW distance is experimentally studied
> in Appendix C.3,
> particularly with respect to the reference $\xi$
> and the numerical discretization of these functions.
> Furthermore,
> SQW as $\xi$-based SW on
> $\mathcal P_2( \mathcal P_2(\mathbb R)) \subset \mathcal P_2( L_2([0,1]))$
> is compared to other distances in Table 1.
>
> In the finite-dimensional setting,
> which is not the focus of the submission,
> our $\xi$-based SW distance in (4) reduces
> for instance to
> the energy-based SW distance (Nguyen & Ho, 2023)
> or
> the Gaussian variant of SW (Nadjahi et al., 2021).
> Thus, various established SW distances can be expressed as special cases of our
> $\xi$-based SW definition.
>
> **Weakness 2:**
> Notice that STLB can be considered as a limit case of the discretized SQW for $\sigma \to 0$.
> More detailed, the numerical estimator of SQW tends to
> the Gaussian SW on $\mathbb{R}^R$,
> which corresponds to the classical SW estimator employed by STLB.
> On the theoretical side, the construction behind STLB does not work for continuous
> shapes, unlike our approach. This limit case is now briefly pointed out in the new supplementary Section C.2.
> Therefore, we would expect SQW and STLB to be on par.
>
> The computational gains regarding GW, TLB, and AE become more apparent for large point clouds or shapes.
> To highlight this, we added a new experiment to Table 1 based on a dataset with 2000 points per shape. Here, we see a 6x speed-up over TLB and 3x over AE. Moreover, we compute the correlation between our SQW estimates and the TLB on these large-scale point clouds, as shown in our updated Figure 1. Here, we see a very clear relation, i.e., a correlation of 0.99 or higher.
> Our main motivation for the shape classification experiment in Section 5.1 is to outline the potential use of our distance in this setting.
>
> **Question 1:**
> In our experiments, we observe robustness with respect to the number of slicing directions after a certain threshold. To illustrate this, we added the new supplementary section C.5.
>
> In Section C.5.1, we added a new experiment. Here, we analyze the correlation between different dataset distances and the accuracy of a 5-nearest-neighbour MNIST classifier. In particular, we performed this experiment with 1000 and 10000 projections for our DSW and the s-OTDD. In this case, we barely observe any differences between 1000 and 10000 projections. Furthermore, DSW displays stronger correlation than the s-OTDD in both cases. The results are visualized in our new Figure 8.
>
> In Section C.5.2, we extend the experiment from Section 5.2 on dataset distances. Here, we recompute the correlation between our DSW and the OTDD on MNIST for various parameter configurations in our new supplementary Table 3, e.g., with 1000, 10 000, 100 000 projections.
> We observe an increase in correlations for more projections, but this effect remains marginal, with correlation coefficients ranging from ~0.9 to ~0.95.
>
> **Question 2:**
> Thanks for drawing our attention to this reference.
> We incorporated this reference into the introduction
> and give a brief conceptual comparison
> after the introduction of $\mathbf{SW}$ at the end of Section 4,
> where we explain that
> our approach is based on an indirect slicing of the domain
> and the reference on a direct slicing
> with respect to the projection to certain geodesic rays.
>
> **Question 3:**
> We refer to our aforementioned experiment in C.5.1, where we correlate our DSW with the performance of an MNIST classifier. The results in Figure 8 show that our DSW displays similar levels of correlation to the OTDD and stronger ones than the s-OTDD.
> However, a detailed analysis on transfer learning applications remains out of the main scope of our current work.

---

> > ### Comment · Reviewer_HPNJ · 2025-11-21
> >
> > I would like to thank the authors for the effort they have put into addressing my comments and for extending the comparison to include the new related work. While I believe that stronger empirical evidence would further strengthen the paper, I find the theoretical framework to be very sound and elegant, and I recommend that the paper be accepted with high regard.

---

> > > ### Author Response · Authors · 2025-11-21
> > > **Thank you**
> > >
> > > Many thanks for the prompt response! We are very happy about your positive recommendation.

---

### Author Response · Authors · 2025-11-20
**Updated Manuscript**

We thank all reviewers for their time and consideration and we appreciate the very positive feedback! Based on your constructive comments, we updated our manuscript and uploaded a new version, including supplementary sections. In our main text, we

- extended our section on shape classification (Section 5.1, updated Figure 1)
- incorporated all references into our text,
- added additional explanations to our main text,
- extended our conclusion.

In our supplementary material, we

- added a practical guide on parameter selection (Appendix C.2),
- extended our parameter sensitivity studies (Appendices C.3-C.6),
- added an experiment on transfer learning (Appendix C.4),
- provided a small background on patch-based methods (Appendix C.7)

Please note that some browsers require deleting the old manuscript and clearing the cache to successfully download the updated file.

In addition, we posted individual replies to all reviews. We hope that we have addressed all of your concerns.

---

### Author Response · Authors · 2025-12-02
**Rebuttal Summary**

Dear **Area Chair** and **Reviewers**,

We would like to thank all of you for your efforts, especially given the circumstances. Moreover, we are happy about the very positive feedback from all reviewers during the initial review process.

Our manuscript proposes an extension of sliced Wasserstein to arbitrary Banach spaces and employs this construction to accelerate Wasserstein-over-Wasserstein computations via slicing. During the review process, we have incorporated the reviewers' comments by extending our experiments, adding various parameter studies, adding a transfer learning experiment, and revising our manuscript accordingly.

In their answers, all reviewers confirmed that we have addressed their concerns. Moreover, the reviewers **HPNJ** and **qqXT** raised their scores, resulting in a **score of 8-8-8-8** (before the score reset).

Best regards

The Authors

---

### Meta-Review · Area_Chair_d2EN · 2025-12-05

**Summary:**

This paper addresses the high computational cost of comparing distributions of distributions (i.e., Wasserstein over Wasserstein (WoW)). The authors propose the Double-Sliced Wasserstein (DSW) distance, a scalable alternative using two levels of slicing: first projecting each distribution to one dimension, then slicing again within the space of quantile functions using Gaussian-based directions. DSW is computationally efficient, stable in high dimensions, and theoretically consistent with WoW for discretized meta-measures. Experiments on 3D shapes, point clouds, and image datasets show that DSW matches or outperforms WoW and sliced-OT approaches, offering a practical tool for hierarchical OT.

Overall, the reviewers are satisfied with the paper, and the authors have provided clear responses during the rebuttal. Therefore, I would also recommend acceptance.

**Reviewer Scores:**

Most of reviewers are happy with the paper in the initial review. One of the reviewers, who is an expert of OT research, promised to increase the review to 8.

---

### Decision · Program_Chairs · 2026-01-26

Accept (Poster)